# Compositional Semantic Parsing with Large Language Models

**Andrew Drozdov**[1,2,*]   **Nathanael Schärli**[1,*]   **Ekin Akyürek**[1,3]   **Nathan Scales**[1]
**Xinying Song**[1]   **Xinyun Chen**[1]   **Olivier Bousquet**[1]   **Denny Zhou**[1]
[1]Google Research   [2]UMass Amherst CICS   [3]MIT CSAIL   [*]Equal contribution

## Abstract

Humans can reason compositionally when presented with new tasks. Previous research shows that appropriate prompting techniques enable large language models (LLMs) to solve artificial compositional generalization tasks such as SCAN. In this work, we identify additional challenges in more realistic semantic parsing tasks with larger vocabulary and refine these prompting techniques to address them. Our best method is based on least-to-most prompting: it decomposes the problem using prompting-based syntactic parsing, then uses this decomposition to select appropriate exemplars and to sequentially generate the semantic parse. This method allows us to set a new state of the art for CFQ while requiring only 1% of the training data used by traditional approaches. Due to the general nature of our approach, we expect similar efforts will lead to new results in other tasks and domains, especially for knowledge-intensive applications.

## 1 Introduction

Compositionality is a key part of human intelligence as it allows us to understand and produce a potentially infinite number of novel combinations of known components (Chomsky, 1957; Montague, 1970; Lake et al., 2017). In contrast, standard neural sequence models, transformers and recurrent neural networks, often fail to capture the compositional structure of the problem domain and thus fail to generalize compositionally (Keysers et al., 2020; Furrer et al., 2020).

Prior efforts to improve compositional generalization primarily rely on specialized architectures or training procedures (Lake, 2019; Chen et al., 2020; Nye et al., 2020; Andreas, 2020; Conklin et al., 2021; Akyürek et al., 2021; Liu et al., 2021). Although effective, these can be task-specific. Even more general purpose methods that rely on data augmentation are limited in the class of data it can support (Shaw et al., 2021; Qiu et al., 2022a). Prompting on the other hand is sufficiently flexible and, with recent advancement of large-scale pretrained language models (LLMs), has become an effective and generic approach to address a wide range of language understanding problems (Brown et al., 2020). Prompting now performs on-par or better than model finetuning in many cases (Wei et al., 2022b; Chowdhery et al., 2022; Wei et al., 2022a; Kojima et al., 2022; Ahn et al., 2022), and might be suitable for improving language model performance on compositional generalization.

In particular, recent work (Zhou et al., 2022) found that least-to-most prompting shows a lot of potential for adapting LLMs for compositional generalization, achieving 99.7% accuracy on SCAN, a commonly used compositional generalization benchmark. Least-to-most prompting decomposes each problem into a series of subproblems, then sequentially solves one after another. However, SCAN is an artificial task built upon a synthetic language with a tiny vocabulary and is generated from a small set of grammar rules, and it is unclear whether strong results transfer to more realistic tasks that are based on a larger vocabulary and more complicated grammars (Furrer et al., 2020).

Additional challenges arise when applying least-to-most prompting to more realistic semantic parsing benchmarks. Among others, they may require information beyond what fits in a single prompt. Also, decomposing a problem is more difficult than with SCAN, exacerbated by constituents that cannot be translated independent of their context. We address these challenges with *dynamic least-to-most prompting*, a generic refinement of least-to-most prompting that involves the following steps: (1) tree-structured decomposition of natural language inputs through LM-predicted syntactic parsing, (2) using the decomposition to dynamically select exemplars, and (3) linearizing the decomposition tree and prompting the model to sequentially generate answers to subproblems.

We evaluate our approach on two realistic benchmarks that, like SCAN, are designed to measure compositional generalization: CFQ (Keysers et al., 2020) and COGS (Kim & Linzen, 2020). On CFQ, our best performing method outperforms previous fully supervised finetuning approaches and achieves a new state-of-the-art accuracy of 95% (averaged across MCD splits) and thereby reduced the error rate by about 45% compared to the previous best result while using about 1% of the training data as candidates for exemplars. On COGS, our approach scores an accuracy of 99.2% when evaluated on the generalization test set, comparable with strong baselines. We also demonstrate robustness of our approach to exemplar pool size, and when using less than 0.1% of the training data as exemplars, *dynamic least-to-most prompting* is still competitive with previous approaches.

## 2 BACKGROUND AND MOTIVATION

### 2.1 COMPOSITIONAL GENERALIZATION

*Compositionality is the idea that the meanings of complex expressions are constructed from the meanings of the less complex expressions that are their constituents. —Fodor & Lepore (2002)*

Given the knowledge of conceptual primitives and a few combinations, *compositional generalization* is the capability to use and comprehend unseen combinations. SCAN (Lake & Baroni, 2018; Loula et al., 2018) is one of the earliest benchmarks that shows neural sequence models cannot systematically generalize to novel combinations of the primitive items of the language. The benchmark requires the learner to translate simple commands to action sequences, where all commands are generated from a set of 20 grammar rules and use a a vocabulary consisting of about 20 words.

Recent work has achieved perfect generalization accuracy on SCAN by inferring grammar rules in symbolic form (Chen et al., 2020; Nye et al., 2020; Liu et al., 2020; Shaw et al., 2021). Most recently, Zhou et al. (2022) demonstrate that SCAN can be solved by least-to-most prompting, which leverages a pretrained large language model (LLM) and a prompt consisting of only 14 exemplars, which is less than 0.1% of the training data used by previous approaches.

### 2.2 LEAST-TO-MOST PROMPTING ENABLES COMPOSITIONAL GENERALIZATION

Least-to-most prompting teaches a language model how to solve a complex problem by reducing it to a set of easier subproblems. This is done by constructing two types of prompts. The first type of prompt tells the language model how to decompose a problem into a list of subproblems, while the second type of prompt describes how to sequentially solve the subproblems.

As an illustration, consider the application of least-to-most prompting to SCAN. The decomposition of the input "look around right thrice and walk twice" yields the following subproblems: "look right", "look around right", "look around right thrice", and "walk twice". Since SCAN commands are generated by a simple grammar of only 20 rules, this decomposition task can be performed using a prompt consisting of only 8 decomposition exemplars.

This decomposition allows the translation of the original input to be produced sequentially rather than in one step (as would be the case with naive prompting). The first subproblem is translated by passing the language model a prompt context consisting of 14 simple translation exemplars followed by the command "look right". The model's answer is then appended to the prompt such that it is used as additional context when translating the next subproblem "look around right", etc.

**Did M1 star M2 , star M3 , and star a art director and editor of M0**
```
SELECT count(*) WHERE { ?x0 edited M0 .  ?x0 art_directed M0 .
   M1 starred ?x0 .  M1 starred M2 .  M1 starred M3 }
```

**What was produced by a art director that M1 and M2 employed**
```
SELECT DISTINCT WHERE { ?x0 produced_by ?x1 .  ?x1 a art_director .
   M0 employed ?x1 .  M1 employed ?x1 }
```

Figure 1: An example of semantic parsing problems in CFQ, where the input is a sentence and the output is its formal representation as a SPARQL query.

## 2.3 LEAST-TO-MOST PROMPTING: LIMITATIONS AND CHALLENGES

While the performance of least-to-most prompting on SCAN is impressive, it is not clear whether and how the same technique can be applied to compositional generalization problems that are based on a more realistic subset of natural language. In particular, we identified three challenges that are common for more realistic natural language tasks and need to be addressed in order to apply least-to-most prompting: (1) decomposition is more challenging, (2) the knowledge required for translation may be too large to fit into a single prompt, and (3) translation of constituents is context-dependent.

As we consider extending least-to-most to the more realistic data setting, we have two semantic parsing benchmarks in mind: CFQ (Keysers et al., 2020) and COGS (Kim & Linzen, 2020).

CFQ is a compositional semantic parsing task where natural language questions need to be translated into SPARQL commands. Compared to SCAN, CFQ is based on a much larger vocabulary as well as more complex linguistic structures produced by a unification-based grammar (Shieber, 2003). As a result, CFQ has proven to be quite challenging for generic ML architectures such as transformers and LSTMs. Even with custom architectures, the best generalization accuracy is less than 91%, achieved by specialized architectures for compositional grammar learning (Liu et al., 2021)

COGS is another semantic parsing task where natural language sentences need to be translated into a formal representation. As for CFQ and SCAN, the training data for COGS contains multiple systematic gaps that can only be addressed by compositional generalization; these include new combinations of familiar syntactic structures and familiar structures. While COGS proves to be quite challenging for generic ML architectures, the best specialized models achieve 99.7% accuracy (Qiu et al., 2022a).

**Natural Language is Challenging to Decompose**  SCAN commands are constructed from eight distinct symbols with a fixed precedence ("left", "right", "twice", "thrice", "opposite", "around", "and", and "after"). Roughly, the decomposition of a SCAN statement resembles that of a mathematical expression with standard arithmetic operations. In practice, the decomposition for SCAN can be predicted by a language model using a simple prompt.

CFQ and COGS sentences represent a richer subset of natural language, meaning the various components and their interactions involve grammatical features such as different parts of speech, grammatical voice, conjunctions, and pronouns. This makes decomposition much more challenging as it requires deep understanding of the underlying linguistic structures.

**Single Prompt Insufficient to Represent Full Label Space**  In the case of SCAN, the knowledge needed to translate a command into a sequence of actions is small enough that it can be captured with about a dozen examples. This is not the case for more realistic semantic parsing problems. For example, CFQ uses more than 50 different Freebase types and relations, and we cannot really expect the model to know the names of those relations without seeing them used in examples. Similarly, COGS uses hundreds of verbs, and their translation depends on details such as whether or not a verb is unaccusative or unergative, which are difficult or impossible to determine without seeing corresponding translation examples.

**Constituent Translation is Context-Dependent**  Least-to-most prompting has only been applied to domains where constituents can be translated independent of their context. The context-free nature of those tasks enables smooth derivation of a final output from the solutions to the subproblems. As an illustration of the context-free nature of SCAN, consider the expression "walk twice", which always translates to "WALK WALK". The constituents in CFQ cannot be translated independent of their context. This is exemplified by the two sentences in Figure 1. In one, the expression "a art director" translates to "?x0 art_directed M0", but translates to "?x1 a art_director" in the other.

As we will detail in the next section, this means that we cannot use the traditional approach for least-to-most prompting, where we first ask the model to translate each subproblem in isolation. Instead, we need to make sure that the subproblems are provided to the model with enough context.

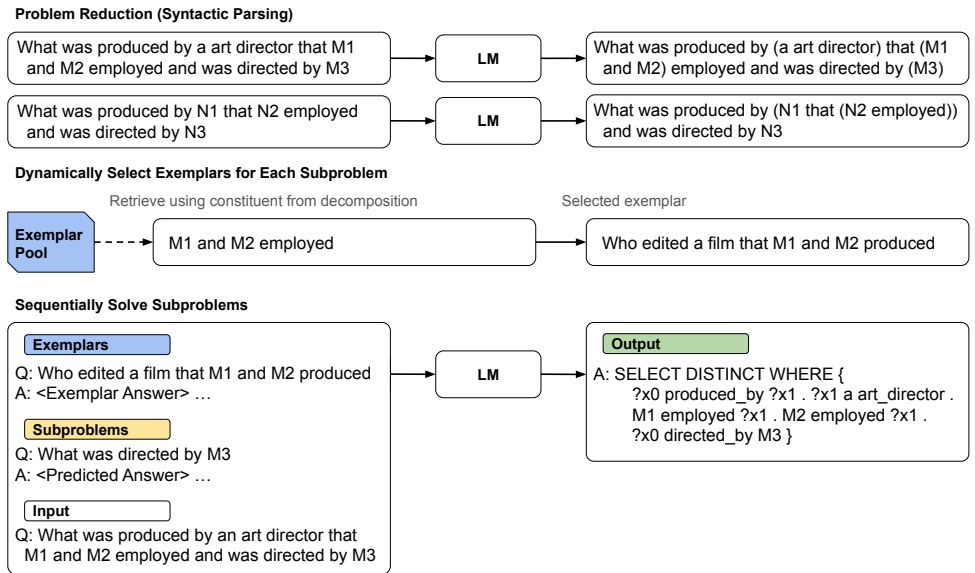

Figure 2: Our application of least-to-most is similar to Zhou et al. (2022) with the differences that we obtain the "problem reduction" via a multi-step syntactic parse of the input, and we dynamically select exemplars from a fixed pool such that they collectively demonstrate as many parts of the decomposition as possible.

# 3 DYNAMIC LEAST-TO-MOST PROMPTING

In this section, we introduce *dynamic least-to-most prompting*, which is an extension of least to most prompting that allows us to overcome the challenges stated above and consequently apply least-to-most prompting to more realistic natural language tasks.

We start by giving a high-level summary of this approach, which is outlined in Figure 2.

1. **Decomposition using LM-based syntactic parsing.** We use a series of prompts to teach the language model to perform a syntactic parse of all possible input sentences. This provides us with a *tree-based decomposition* rather than a linear decomposition obtained by traditional least-to-most prompting.

2. **Dynamic selection of exemplars based on the decomposition.** We sample a small subset of the training set as a pool of candidate exemplars. For each new input sentence to process, we dynamically select exemplars from this pool such that they collectively demonstrate relevant knowledge needed to translate the input sentences. This is done by matching the decomposition tree of the input against the decomposition tree of the candidate exemplars.

3. **Sequential solution based on the decomposition.** We use the tree-based decomposition of the input sentence to generate a linear sequence of other relevant simpler sentences. We then construct a prompt including the dynamically selected exemplars and use it to sequentially predict the solutions for the simpler sentences before generating the final output.

## 3.1 DECOMPOSITION USING LM-BASED SYNTACTIC PARSING

As discussed in Section 2.3, decomposition is more challenging for realistic tasks such as CFQ and COGS than for artificial tasks like SCAN. Indeed, while decomposing SCAN commands is similar to decomposing mathematical expressions with standard arithmetic operations, decomposing sentences corresponding to more realistic subsets of natural language essentially becomes a problem of syntactic parsing. We find it natural to decompose problems using a tree structure guided by that syntax (see Figure 3).

To teach LMs to perform decomposition, we divide the syntactic parsing task into multiple steps, such as subclause identification, noun phrase identification, verb phrase identification, phrase an-

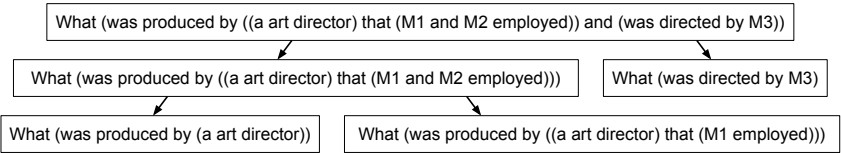

Figure 3: Syntactic parse of a CFQ input and its decomposition into subproblems. Like the input, the subproblems are well-formed sentences.

notation, and verb normalization. For each step, we provide the LM with exemplars that illustrate the task to be performed.[1,2] See Appendix A.1 and Appendix B.1 for detailed description of the decomposition process and prompts used for CFQ and COGS, respectively.

## 3.2 DYNAMIC EXEMPLAR SELECTION

Prompt size is limited, so rather than attempt to represent all relevant knowledge in single prompt, for each input we dynamically select a set of relevant exemplars from a pre-selected exemplar pool.

**Choosing the exemplar pool.**    The exemplar pool is typically a small subset of the available training data. The knowledge required for CFQ is rich and diverse, so we randomly sample 1000 exemplars from the training data for this purpose (separately for each split). For COGS, it is important to include in the exemplar pool translations of as many different verbs and verb phrases as possible. Therefore, we selected from the training data one exemplar for each unergative and unaccusative verb, as well as 3 examples for each type of verb phrase (e.g., active, passive, with and without recipient, etc.). This resulted in a relatively small exemplar pool consisting of only 89 exemplars (which is used in addition to a static context consisting of 28 exemplars).

**Selecting exemplars.**    The goal of exemplar selection is to provide the LLM with the most relevant information needed to process a given input sentence. We do this by making sure that as many nodes as possible of the decomposition tree of the input are covered by the decomposition trees of the selected exemplars. Specifically, we perform the following bottom-up and top-down matching.

- *Top-down matching:* We begin by anonymizing the decomposition tree of the input. For instance, the example "What was produced by a art director that M1 and M2 employed" is anonymized to "What V N that M and M V", where V stands for verb, N for noun, and M for entity. Starting at the top of the anonymized tree, we use a heuristic approach to find exemplars such that all nodes are covered, prioritizing exemplars that match large subtrees.

- *Bottom-up matching*: Then, we try to make sure that all leaf phrases are covered by an exemplar. If there is more than one exemplar for a certain phrase, we prefer exemplars where the phrase occurs within a similar anonymized subtree. For instance, for the phrase "M1 and M2 employed" (which anonymizes to "M and M V"), we would prefer an exemplar containing "M1 and M2 produced" over both "art director employed" and "directed by M1 and M2".

Depending on its complexity, we select for each input between 4 and 35 exemplars for CFQ and between 1 and 3 exemplars for COGS. Full details of the anonymized tree and our heuristic matching algorithms are in Appendix A.2 for CFQ and Appendix B.2 for COGS.

---

[1]To better handle compositionality of the input sentences, some prompts may be applied iteratively. For instance, some sentences in COGS have up to 12 nested that-clauses. Therefore, we use a prompt that extracts the outer-most that-clause and apply this prompt until all those clauses are identified.

[2]Because of a lack of golden data, we did not directly evaluate syntactic parsing. However, a manual inspection reveals desirable outputs for both CFQ and COGS, which speaks for the ability of LMs to perform syntactic parsing when the task is broken down into individual steps that are illustrated with appropriate prompts.

**Chain-of-Thought**

> **Input**
> Q: What was produced by an art director that
>    M1 employed and was directed by M3
>
> **Intermediate Steps**
> Type: what => DISTINCT
> 1. There was an art director (?x1) that produced ?x0 =>
>    ?x1 a art_director, ?x0 produced_by ?x1
> 2. ?x1 is employed by M1 => M1 employed ?x1
> 3. ?x0 was directed by M3 => ?x0 directed_by M3
>
> **Output**
> A: SELECT DISTINCT WHERE {
>    ?x0 produced_by ?x1 . ?x1 a art_director .
>    M1 employed ?x1 . ?x0 directed_by M3 }

**Dynamic Least-to-Most**

> **Subproblems**
> Q: What was directed by M3
> A: <Predicted Answer>
> Q: What was produced by an art director
> A: <Predicted Answer>
> …
> **Input**
> Q: What was produced by an art director that
>    M1 employed and was directed by M3
>
> **Output**
> A: SELECT DISTINCT WHERE {
>    ?x0 produced_by ?x1 . ?x1 a art_director .
>    M1 employed ?x1 . ?x0 directed_by M3 }

Figure 4: Prompt designs for semantic parsing. Chain-of-thought (left) generates intermediate steps before the final output. Dynamic least-to-most (right) first sequentially predicts solutions to subproblems before generating the final output—the subproblems are extracted through a separate prompt.

## 3.3 Sequential solution

This is similar to the solution step for traditional least-to-most prompting. The main difference is that we cannot translate the constituents in isolation because they might not be well-formed sentences and their translation is context-dependent. Instead, we linearize the decomposition tree into a sequence of increasingly complex subproblems and sequentially predict their solutions.

In Figure 4, we show a barebones snapshot of dynamic least-to-most prompting, immediately before predicting the final output. In previous steps, the model predicted via prompting the solution to the first subproblem, "What was directed by M3", then appended the prediction to the prompt before proceeding with, "What was produced by an art director", etc. Not displayed in the figure are dynamically selected exemplars and a fixed list of partial exemplars that we prepend to each prompt.[3]

## 4 Baseline: Prompting without Decomposition

To demonstrate the effectiveness of dynamic least-to-most prompting, we compare against a strong prompting baseline called chain-of-thought prompting (Wei et al., 2022b). Chain-of-thought generates intermediate steps before predicting the final answer and has been shown to improve reasoning capabilities of language models. Unlike least-to-most prompting, chain-of-thought does not have a decomposition step, potentially limiting its effectiveness for compositional generalization tasks.

### 4.1 Chain-of-Thought Prompt Design

Our chain-of-thought prompt is shown in Figure 4. It first categorizes the query, then generates quasi-alignments between the text and output statements. This represents synchronicity in the same spirit as other higher performing semantic parsers do (Qiu et al., 2022a), but the intermediate constituents and clauses need not map exactly to what is seen in the input and output. The flexibility of chain-of-thought is convenient for data like CFQ where inclusion of variables is not compatible with synchronous grammar training and inference (Wong & Mooney, 2007).

### 4.2 Dynamically selecting exemplars based on lexical similarity

Since there is no decomposition with chain-of-thought, we rank exemplars by bag-of-words similarity with the current sentence. To reduce redundancy, we select exemplars one by one, and at each iteration prioritize exemplars with relevant words that have not been found yet. This approach is not deterministic—i.e., selecting different exemplars in the first iteration can alter results. In practice, when using chain-of-thought we find it is helpful to sample multiple exemplar lists, then use

---

[3]A partial exemplar is generic and represents fundamental language phenomena (see Appendix A.3.1). Complete details of the prompts for dynamic least-to-most are in Appendices A.3 (CFQ) and B.3 (COGS).

temperature-based decoding to sample multiple predictions per list, and finally aggregate predictions by plurality vote using self-consistency (Wang et al., 2022; Shi et al., 2022; Li et al., 2022).

# 5 EXPERIMENTS AND RESULTS

## 5.1 DATA

**Datasets.** We empirically measure the effectiveness of prompting on two semantic parsing datasets. CFQ (Keysers et al., 2020) has three maximum compound divergence splits (MCD1, MCD2, MCD3) for measuring compositional generalization, each with 95743/11968/11968 sentences in their train/validation/test splits. COGS (Kim & Linzen, 2020) has 24155/3000/21000 sentences in their train/validation/generalization splits.

**Exemplar pools.** Also described in Section 3.2. For each CFQ split, we sampled 1000 training examples to use as potential exemplars (about 1% of the training data). For COGS, we manually selected 89 training examples as potential exemplars (about 0.4% of the training data).

**Preprocessing.** For CFQ we replace freebase identifiers with human-readable strings.[4] We also remove clauses that are redundant from a prediction perspective, which always appear alongside the same properties in both train and evaluation data. For COGS we use the variable-free and equivalent outputs introduced by Qiu et al. (2022a). See Appendix D for details.

**Evaluation.** We measure accuracy using exact match (EM). This is computed as an exact string match between ground truth and predict labels, and no partial credit is assigned. To make this metric interpretable for CFQ we apply normalization to outputs, including sorting properties and applying a deterministic argument ordering (additional details in Appendix D.1.2).[5] For COGS we add any missing closing parentheses at the end of the output before computing EM.

## 5.2 EXPERIMENTAL DESIGN CHOICES

The number of exemplars used in dynamic least-to-most varies based on the complexity of the decomposition tree, as does the number of subproblems. Predictions are made with greedy decoding. Since there is a single final output, no self-consistency is needed.

To improve performance with vanilla few-shot prompts (simple input/output exemplars) and chain-of-thought prompts, we sample $n = 4$ different exemplar lists (each consisting of $k = 15$ exemplars), then sample $s = 4$ outputs per list using temperature-based decoding, yielding $n \cdot s = 16$ outputs that are aggregated with self-consistency.[6]

We use `code-davinci-002` hosted by OpenAI for all experiments described in this paper. This is a version of InstructGPT (Ouyang et al., 2022) finetuned on code, referred to as Codex.[7] Hyperparameters are summarized in Appendix D.3.

## 5.3 RESULTS

**CFQ** Our main results are on CFQ test splits and are reported in Table 1. Not only does this show that prompting enables compositional generalization on a realistic natural language task, dynamic least-to-most sets a new state of the art (95.0% accuracy) while only using about 1% of the training data, whereas traditional approaches are fully supervised.

---

[4]We manually mapped 52 properties to a human-readable form, which took about one hour to complete. This heavily relies on the original ID, and results in properties that are more feasible to predict. For example, mapping `ns:film.film_art_director.films_art_directed` to `art_directed`.

[5]Any specialized preprocessing and evaluation steps we take for CFQ are tailored for prompting and are not necessary for fully supervised training. In order to verify this, we reproduced experiments with T5-base using our same preprocessing and evaluation. The results match previously published exact match within 1-2 points on MCD1, MCD2, and MCD3. COGS does not require such steps.

[6]Vanilla few-shot and chain-of-thought do worse without self-consistency (see results in Appendix E.3).

[7]We use a code-based model because it outperformed text-based GPT-3 in early experiments. Similar observations have been made in prior work on semantic parsing (Shin et al., 2021; Shin & Van Durme, 2022).

|  | MCD1 | MCD2 | MCD3 | Ave. |
|---|---|---|---|---|
| **Fully Supervised** | | | | |
| T5-base | 58.5 | 27.0 | 18.4 | 34.6 |
| T5-large | 65.1 | 32.3 | 25.4 | 40.9 |
| T5-3B | 65.0 | 41.0 | 42.6 | 49.5 |
| HPD | 79.6 | 59.6 | 67.8 | 69.0 |
| T5-base + IR | 85.8 | 64.0 | 53.6 | 67.8 |
| T5-large + IR | 88.6 | 79.2 | 72.7 | 80.2 |
| T5-3B + IR | 88.4 | 85.3 | 77.9 | 83.9 |
| LeAR | 91.7 | 89.2 | 91.7 | 90.9 |
| **Prompting** | | | | |
| (Ours) Dyn. L2M | **94.3** | **95.3** | **95.5** | **95.0** |

Table 1: Test accuracy across the MCD splits for the CFQ dataset. T5: Herzig et al. (2021), HPD: Guo et al. (2020), LeAR: Liu et al. (2021).

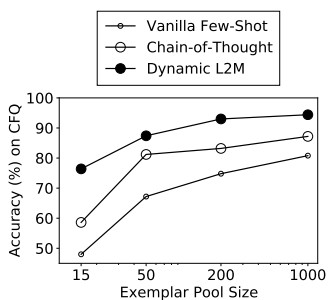

Figure 5: Comparing prompting methods and ablating across exemplar pool size.

|  | Gen. |
|---|---|
| **Fully Supervised** | |
| LeAR (Liu et al., 2021) | 97.7 |
| T5-base (Qiu et al., 2022a) | 89.8 |
| T5-base + CSL (Qiu et al., 2022a) | **99.5** |
| **Prompting** | |
| (Ours) Dynamic Least-to-Most | 99.2 |

Table 2: Accuracy on COGS generalization set. The COGS data is not SQL-like, and has a more diverse lexicon compared with CFQ.

|  | MCD1 |
|---|---|
| Dynamic Least-to-Most | 94.6 |
| Using BoW Exemplars | 92.2 |
| Using Constant Exemplars | 76.4 |
| 50% Decomposition | 88.8 |
| 2-Step Decomposition | 83.2 |
| No Decomposition | 75.2 |

Table 3: Ablating exemplar selection and decomposition for dynamic least-to-most.

**COGS** We run experiments on COGS with minimal changes to our approach (Table 2) even though the output space is quite different from CFQ. Dynamic least-to-most prompting scores 99.2% accuracy (using 0.4% of the training data), reinforcing the generic nature of our approach.

## 6 DISCUSSION AND ANALYSIS

**Comparison of prompting methods** In Figure 5, we compare dynamic least-to-most (L2M) with vanilla few-shot prompting and chain-of-thought prompting (CoT) enhanced with self-consistency. We measure performance on a random 500-sentence subset of CFQ's MCD1 test split. CoT (87.2%) outperforms vanilla (80.8%) and most baselines when extrapolating to the full data, but L2M (94.6%) achieves the best result.

L2M is both more accurate and more efficient than other prompting methods. L2M performs favorably across different sizes of the exemplar pool. It is also 2x as fast as CoT despite its sequential nature, since CoT uses self-consistency across multiple exemplar lists and outputs. Without self-consistency, CoT accuracy drops from 87.2% to 75.4% and vanilla drops from 80.8% to 69.8%.

**Exemplar selection and question decomposition** In Table 3, we ablate configurations for dynamic least-to-most. We measure performance on a random 500-sentence subset of CFQ's MCD1 test split. Performance drops from 94.6% to 92.2% when selecting exemplars according to our bag-of-words (BoW) algorithm or to 76.4% when using a fixed set of exemplars for all examples.

Both exemplar selection and question decomposition are crucial for the best L2M performance. L2M uses a different number of decomposition steps per example. If we reduce the amount of decomposition to half as many steps then performance drops to 88.8%, or to 83.2% when doing at most two steps. If we skip decomposition entirely then performance drops further to 75.2%. See Appendix E.2 for more ablations together with a qualitative analysis of the associated errors.

**Risk of memorization** For pretrained models, leakage of the test data is a potential concern (Krishna et al., 2020; Carlini et al., 2021). However, given the inferior performance of vanilla few-shot prompting, we attribute success of prompting to in-context learning rather than memorization. Furthermore, least-to-most prompting requires the LM to translate as intermediate steps new subproblems that are guaranteed to be unseen during training.

# 7 RELATED WORK

**Compositional generalization** Compositional generalization in machine learning has been a challenging problem that attracts attention across fields, including vision (Johnson et al., 2017; Bahdanau et al., 2019; Ruis et al., 2020; Nikolaus et al., 2019) and language domains (Lake & Baroni, 2018; Keysers et al., 2020; Kim & Linzen, 2020; Shaw et al., 2021; Yin et al., 2021; Gan et al., 2022). A number of approaches have been proposed to improve compositional generalization on SCAN (Lake & Baroni, 2018; Loula et al., 2018), including specialized design of neural model architectures (Chen et al., 2020; Nye et al., 2020; Liu et al., 2020; Shaw et al., 2021; Russin et al., 2019; Li et al., 2019; Gordon et al., 2020; Herzig & Berant, 2021) and training algorithms (Lake, 2019; Kim, 2021), training data augmentation (Andreas, 2020; Akyürek et al., 2021), and prompting (Zhou et al., 2022). While 100% accuracy has been accomplished on SCAN (Chen et al., 2020; Nye et al., 2020; Liu et al., 2020; Shaw et al., 2021), good performance on SCAN does not necessarily transfer to more challenging compositional generalization problems (Furrer et al., 2020). Notably, although least-to-most prompting has achieved 99.7% accuracy on SCAN (Zhou et al., 2022), prior attempts on prompting for semantic parsing still demonstrate limited compositional generalization performance (Qiu et al., 2022b). In this work, we propose prompting schemes to bridge this gap.

To improve compositional generalization for semantic parsing, recent works incorporate a latent syntactic component (Qiu et al., 2022a; Liu et al., 2021). Similarly to symbolic grammar learning techniques on SCAN, these approaches achieve impressive performance on several benchmarks, and represent the previous state of the art on CFQ (Keysers et al., 2020) and COGS (Kim & Linzen, 2020). Other lines of work improve the performance on CFQ through specialized decoding algorithms, including graph decoding (Gai et al., 2021) and hierarchical poset decoding (Guo et al., 2020). Yet others exploit correlations between the input and output tokens, e.g. through loss functions with attention supervision (Yin et al., 2021), using a lexicon (Akyürek & Andreas, 2021), and reformulating the label space (Herzig et al., 2021). Without relying on specialized model architectures or training algorithms, our results demonstrate generic prompting schemes based on decomposition demonstrate strong results on CFQ and COGS, and achieve state-of-the-art results on CFQ.

**Prompting** The most similar work to ours is SeqZero (Yang et al., 2022), but there are key differences. SeqZero decomposes semantic parsing into generating three parts separately (SELECT, FROM, and WHERE parts of the output), and further decomposes WHERE into generating each clause separately. This means that decomposition is conducted via a fixed, rule-based system. In contrast, our decomposition is automatically accomplished by prompting. We use the syntactic parse of the sentence to create different related sentences, such as by simplifying conjunctions or removing text fragments. This is more general than SeqZero and it is readily applicable to many natural language tasks. For example, we successfully used our approach on COGS even though the output does not resemble SQL. Furthermore, SeqZero is an ensemble of a finetuned BART and a zero-shot model, while we use only prompting with large language models and forego finetuning entirely.

# 8 CONCLUSION

Through dynamic least-to-most prompting, we demonstrate state-of-the-art performance on a difficult natural language semantic parsing benchmark that measures compositional generalization. Our results are achieved using about 1% of the training data from traditional finetuning approaches. Many machine learning models struggle with compositional generalization, and our findings should facilitate future research that enables this capability through task decomposition. We expect dynamic least-to-most prompting to have immediate impact in a variety of settings. It is flexible and general purpose, enabling quick adaptation to new tasks and domains, especially for knowledge-intensive applications of language models, where precise semantic parsing can be directly leveraged.

REPRODUCIBILITY STATEMENT

Throughout our work we aim to provide exhaustive details about prompt design and exemplar selection, and we include all the prompts we use in the Appendix. To ease future use, we further outline key details related to reproducibility in Appendix F.

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

# Appendix

## Table of Contents

**Which film editor was influenced by a cinematographer that wrote M3 , M4 , and M5 and influenced by M1 's editor**

```
SELECT DISTINCT ?x0 WHERE { ?x0 a film_editor .  ?x0 influenced_by ?x1 .
    ?x0 influenced_by ?x2 .  ?x1 edited M1 .  ?x2 a cinematographer .
    ?x2 wrote M3 .  ?x2 wrote M4 .  ?x2 wrote M5 }
```

Figure 6: A CFQ example, which we use to illustrate the application of dynamic least-to-most prompting to CFQ.

# A  LEAST-TO-MOST PROMPTING FOR CFQ

In this section, we provide more details on the application of dynamic least-to-most prompting for CFQ. In particular, we detail all the prompts and show how the example in Figure 6 is processed step by step.

## A.1  CFQ DECOMPOSITION: DETAILS AND PROMPTS

As discussed in Section 3.1, we use prompting-based syntactic parsing to decompose CFQ questions. To teach LMs to perform this kind of decomposition, we divide the syntactic parsing task into the following steps:

1. Noun phrase identification
2. Subclause identification
3. Verb and other phrase identification
4. Part-of-speech tagging and phrase labeling
5. Verb normalization

Consider the question from Figure 6.

- Which film editor was influenced by a cinematographer that wrote M3 , M4 , and M5 and influenced by M1 's editor

The first step, noun phrase identification, yields:

- Which (film editor) was influenced by (a cinematographer) that wrote (M3 , M4 , and M5) and influenced by (M1 's editor)

Then, we replace the identified noun phrases with placeholders N1, N2, N3, and N4, which yields:

- Which N1 was influenced by N2 that wrote N3 and influenced by N4

The next step, subclause identification, uses the form with placeholders and yields:

- Which N1 was influenced by ((N2) that (wrote N3)) and influenced by N4

Then we have another round of applying placeholders, on the subclause this time:

- Which N1 was influenced by N5 and influenced by N4

The third step is to identify verb phrases and other miscellaneous phrases, which yields:

- (Which) (N1) ((was influenced by N5) and (influenced by N4))
- that (wrote N3)

As a fourth step, we perform various part-of-speech tagging and phrase labeling. We start with the noun phrases, which yields:

- N=(film editor)
- [a] N=(cinematographer)
- M=(M3 , M4 , [and], M5)
- M=(M1) P=('s) N=(editor)

We continue with verb phrases, which yields:

- W=(Which)
- V=([was] influenced by) (N5)
- V=(influenced by) (N4)
- V=(wrote) (N3)

As a fifth step, we normalize all the verbs: "was" yields "be", "influenced" yields "influence", and "wrote" yields "write". Finally, we run a few lines of Python code that puts all these parts back together to obtain a parse tree. Note that the normalized verbs do not replace the original ones but are kept in addition, which allows us to obtain higher recall when selecting exemplars.

This yields the following fully decomposed and annotated question:

- W=(Which) N=(film editor) ((V=([was] influenced by) ([a] N=(cinematographer) that (V=(wrote) N=(M3 , M4 , [and] M5)))) and (V=(influenced by) (M=(M1) P=('s) N=(editor))))

Below, we present the exact prompt contexts used for each of the parsing steps.

### A.1.1 STEP 1: NOUN PHRASE IDENTIFICATION

Q: Was M0 's producer a art director 's sibling
A: Was (M0 's producer) (a art director 's sibling)

Q: Did M1 influence a French producer and editor 's child 's spouse
A: Did (M1) influence (a French producer and editor 's child 's spouse)

Q: Which female French costume designer , producer , and editor of M3 and M4 was M5 's parent
A: Which (female French costume designer , producer , and editor of M3 and M4) was (M5 's parent)

Q: Was a Dutch parent and spouse a editor , producer , and writer of a film 's prequel
A: Was (a Dutch parent and spouse) (a editor , producer , and writer of a film 's prequel)

Q: Who employed , met , and was influenced by a female film director 's Spanish parent and married M1
A: Who employed , met , and was influenced by (a female film director 's Spanish parent) and married (M1)

Q: Were M3 and M6 executive produced by , written by , and directed by a writer 's spouse , friend , and employer , and produced by a child
A: Were (M3 and M6) executive produced by , written by , and directed by (a writer 's spouse , friend , and employer) , and produced by (a child)

Q: What film director and editor of M0 , M1 , and M2 married , divorced , and was kissed by a Spanish editor and producer of M5
A: What (film director and editor of M0 , M1 , and M2) married , divorced , and was kissed by (a Spanish editor and producer of M5)

Q: Which child of a production company did M0 acquire
A: Which (child of a production company) did (M0) acquire

Q: Which star of M5 was influenced by a person , influenced by M0 , M1 , M2 , and M3 , and influenced by M4
A: Which (star of M5) was influenced by (a person) , influenced by (M0 , M1 , M2 , and M3) , and influenced by (M4)

Q: Was a screenwriter employed by M1 , M2 , and M3 and employed by M4 , M5 , and M6 M7 's spouse
A: Was (a screenwriter) employed by (M1 , M2 , and M3) and employed by (M4 , M5 , and M6) (M7 's spouse)

Q: Was M4 produced by a German screenwriter that M1 and spouse of M2 's sibling were influenced by
A: Was (M4) produced by (a German screenwriter) that (M1 and spouse of M2 's sibling) were influenced by

Q: Was M0 M1 's sibling and spouse
A: Was (M0) (M1 's sibling and spouse)

Q: Which film did M3 's employer 's Mexican employee produce and M1 direct
A: Which (film) did (M3 's employer 's Mexican employee) produce and (M1) direct

Q: Did M3 's parent 's employee influence M0 and M1 and influence M2 's founder and employee
A: Did (M3 's parent 's employee) influence (M0 and M1) and influence (M2 's founder and employee)

Q: What male director of M3 and M4 did M0 and M1 influence
A: What (male director of M3 and M4) did (M0 and M1) influence

Q: Which female person whose sibling was influenced by M3 and was influenced by M4 and M5 directed M2
A: Which (female person) whose (sibling) was influenced by (M3) and was influenced by (M4 and M5) directed (M2)

Q: Did M0 direct , produce , executive produce , edit , and write M1 , M2 , and M3
A: Did (M0) direct , produce , executive produce , edit , and write (M1 , M2 , and M3)

Q: Was M0 executive produced by , edited by , written by , and directed by M1 , M2 , and M3
A: Was (M0) executive produced by , edited by , written by , and directed by (M1 , M2 , and M3)

Q: Who influenced and was influenced by M1 's female actor 's parent
A: Who influenced and was influenced by (M1 's female actor 's parent)

Q: Did M0 's editor , costume designer , star , writer , and art director executive produce and produce M1
A: Did (M0 's editor , costume designer , star , writer , and art director) executive produce and produce (M1)

Q: Which film that was written by M1 M2 directed
A: Which (film) that was written by (M1) (M2) directed

Q: Which director of M3 , M4 , and M5 was a Japanese screenwriter that M1 employed and was founded by
A: Which (director of M3 , M4 , and M5) was (a Japanese screenwriter) that (M1) employed and was founded by

Q: Did M1 's executive producer employ M2 , edit M3 , employ a film director , and employ M4
A: Did (M1 's executive producer) employ (M2) , edit (M3) , employ (a film director) , and employ (M4)

Q: Was a film whose star , writer , cinematographer , and executive producer executive produced , edited , and wrote M0 M1
A: Was (a film) whose (star , writer , cinematographer , and executive producer) executive produced , edited , and wrote (M0) (M1)

Q: Was M1 a film that M0 's editor distributed
A: Was (M1) (a film) that (M0 's editor) distributed

Q: Was M1 a child of a production company 's parent and child
A: Was (M1) (a child of a production company 's parent and child)

Q: What did a director that M1 influenced and M2 influenced write
A: What did (a director) that (M1) influenced and (M2) influenced write

Q: What was written by M0 's art director and executive producer
A: What was written by (M0 's art director and executive producer)

Q: Did M1 influence a production company , influence M2 , influence M3 , M4 , and M5 , and influence M6
A: Did (M1) influence (a production company) , influence (M2) , influence (M3 , M4 , and M5) , and influence (M6)

### A.1.2 STEP 2: SUBCLAUSE IDENTIFICATION

Q: What N1 that M0 was employed by directed M5
A: What ((N1) that (M0 was employed by)) directed M5

Q: What N1 was N2 that M2 influenced
A: What N1 was ((N2) that (M2 influenced))

Q: Did N1 that N2 married meet N3
A: Did ((N1) that (N2 married)) meet N3

Q: Did M2 marry N1 that M1 was edited by , directed by , and written by
A: Did M2 marry ((N1) that (M1 was edited by , directed by , and written by))

Q: Which N1 that was written by M1 M2 directed
A: Which ((N1) that (was written by M1)) M2 directed

Q: Which N1 that N2 influenced was influenced by and married N3
A: Which ((N1) that (N2 influenced)) was influenced by and married N3

Q: Which director of N1 was N2 that M1 employed and was founded by
A: Which director of N1 was ((N2) that (M1 employed and was founded by))

Q: Was N0 that M1 influenced and M2 was influenced by N1
A: Was ((N0) that (M1 influenced and M2 was influenced by)) N1

Q: What N1 that N2 influenced and N3 was influenced by influenced M1
A: What ((N1) that (N2 influenced and N3 was influenced by)) influenced M1

Q: Who was influenced by N1 that wrote N2 and influenced by N3
A: Who was influenced by ((N1) that (wrote N2)) and influenced by N3

Q: Was M4 produced by N1 that N2 were influenced by
A: Was M4 produced by ((N1) that (N2 were influenced by))

Q: Was M0 N1 that M2 starred and was written by

A: Was M0 ((N1) that (M2 starred and was written by))

Q: Was N1 N2 that M2 employed
A: Was N1 ((N2) that (M2 employed))

Q: Who was N0 that wrote N1 and edited N2
A: Who was ((N0) that (wrote N1 and edited N2))

Q: Did N1 marry and divorce N2 whose N3 wrote M3
A: Did N1 marry and divorce ((N2) whose (N3 wrote M3))

Q: Did N1 whose N2 was employed by M2 and was employed by M3 direct and write M1
A: Did ((N1) whose (N2 was employed by M2 and was employed by M3)) direct and write M1

Q: Did M3 found M4 and found N1 whose N2 wrote M1
A: Did M3 found M4 and found ((N1) whose (N2 wrote M1))

Q: What N1 whose N2 was influenced by N3 played M1
A: What ((N1) whose (N2 was influenced by N3)) played M1

Q: Which N1 whose N2 was influenced by M3 and was influenced by N3 directed M2
A: Which ((N1) whose (N2 was influenced by M3 and was influenced by N3)) directed M2

Q: Which N1 was influenced by N2 whose N3 edited M2 and employed by M3
A: Which N1 was influenced by ((N2) whose (N3 edited M2)) and employed by M3

Q: Was N1 whose N2 executive produced , edited , and wrote M0 M1
A: Was ((N1) whose (N2 executive produced , edited , and wrote M0)) M1

Q: Was M1 N2 whose N3 directed , produced , and wrote N4
A: Was M1 ((N2) whose (N3 directed , produced , and wrote N4))

Q: Was N1 that N2 directed N3
A: Was ((N1) that (N2 directed)) N3

Q: Was N1 whose N2 executive produced M1 and edited M0 M3
A: Was ((N1) whose (N2 executive produced M1 and edited M0)) M3

Q: What N1 was N2 whose N3 wrote N4
A: What N1 was ((N2) whose (N3 wrote N4))

Q: Which N1 that N2 were written by and were produced by was N3
A: Which ((N1) that (N2 were written by and were produced by)) was N3

Q: Was M2 N2 whose N3 was employed by and founded M1
A: Was M2 ((N2) whose (N3 was employed by and founded M1))

Q: Which N1 was N2 whose N3 produced and executive produced M2
A: Which N1 was ((N2) whose (N3 produced and executive produced M2))

Q: Was N0 that N1 were executive produced by , were edited by , were written by , and starred M0
A: Was ((N0) that (N1 were executive produced by , were edited by , were written by , and starred M0))

Q: Which N0 that M2 employed executive produced M1
A: Which ((N0) that (M2 employed)) executive produced M1

Q: Who was N0 that M2 was influenced by and married
A: Who was ((N0) that (M2 was influenced by and married))

Q: Which N0 that M1 was edited by did M2 influence
A: Which ((N0) that (M1 was edited by)) did M2 influence

Q: Were N0 directed by and produced by N1 that executive produced M1
A: Were N0 directed by and produced by ((N1) that (executive produced M1))

Q: What N0 that M1 was executive produced by and written by did N1 marry
A: What ((N0) that (M1 was executive produced by and written by)) did N1 marry

Q: Was N0 that M1 married N1 that directed and edited N2
A: Was ((N0) that (M1 married)) ((N1) that (directed and edited N2))

Q: What N0 influenced by M1 and influenced by N1 that founded N2 edited M2
A: What N0 influenced by M1 and influenced by ((N1) that (founded N2)) edited M2

Q: Was N0 that N1 were edited , art directed , and produced by M0
A: Was ((N0) that (N1 were edited , art directed , and produced by)) M0

### A.1.3 STEP 3: VERB PHRASE IDENTIFICATION

**Verb phrase identification in subclauses**
Q: that N1 influenced and N2 was influenced by
A: that ((N1 influenced) and (N2 was influenced by))

Q: that was edited by N1 and starred N2
A: that ((was edited by N1) and (starred N2))

Q: whose N1 was influenced by M3 and was influenced by N2
A: whose (N1) ((was influenced by M3) and (was influenced by N2))

Q: whose N1 produced and edited M2
A: whose (N1) (produced and edited) (M2)

Q: whose N1 married M2 , met M3 , and played M4
A: whose (N1) ((married M2) , (met M3) , and (met M3))

Q: that M2 was influenced by and M3 influenced
A: that ((M2 was influenced by) and (M3 influenced))

Q: whose N1 executive produced , edited , and wrote M0
A: whose (N1) (executive produced , edited , and wrote) (M0)

Q: whose N1 was influenced by N2
A: whose (N1) (was influenced by) (N2)

Q: that N1 influenced
A: that (N1) (influenced)

Q: that M1 starred and was written by
A: that (M1) (starred and was written by)

Q: that M1 was edited by , directed by , and written by
A: that (M1) (was edited by , directed by , and written by)

Q: whose N1 was employed by M2 and was employed by M3 direct and write M1
A: whose (N1) ((was employed by M2) and (was employed by M3)) (direct and write) (M1)

Q: whose N1 was employed by and founded M1

A: whose (N1) (was employed by and founded) (M1)

Q: that M1 influenced
A: that (M1) (influenced)

Q: that directed and executive produced N1
A: that (directed and executive produced) (N1)

Q: that M2 was influenced by and married
A: that (M2) (was influenced by and married)

Q: that N1 were influenced by , M3 was edited by , and M4 married
A: that ((N1 were influenced by) , (M3 was edited by) , and (M4 married))

Q: that N1 were written by
A: that (N1) (were written by)

Q: that was founded by and employed N2
A: that (was founded by and employed) (N2)

Q: that was influenced by M2 and was employed by M3 and M4
A: that ((was influenced by M2) and (was employed by M3 and M4))

Q: that N1 married
A: that (N1) (married)

Q: that edited N1 and produced M2
A: that ((edited N1) and (produced M2))

Q: whose N1 edited M0 and executive produced N2
A: whose (N1) ((edited M0) and (executive produced N2))

Q: whose N1 edited N2
A: whose (N1) (edited) (N2)

Q: that M1 influenced and employed
A: that (M1) (influenced and employed)

Q: that wrote N1 and edited N2
A: that ((wrote N1) and (edited N2))

Q: that edited and wrote N1
A: that (edited and wrote) (N1)

Q: whose N1 was employed by and founded M1
A: whose (N1) (was employed by and founded) (M1)

Q: whose N1 married M2 and married N2
A: whose (N1) ((married M2) and (married N2))

Q: that played M2 , played M3 , and played M4
A: that ((played M2) , (played M3) , and (played M4))

Q: that wrote , edited , executive produced , and directed N1
A: that (wrote , edited , executive produced , and directed) (N1)

Q: that N3 were written by and art directed by
A: that (N3) (were written by and art directed by)

**Verb phrase identification in main clauses**
Q: Was N1 N2
A: (Was) (N1) (N2)

Q: Did M1 influence N2
A: (Did) (M1) (influence) (N2)

Q: Which N1 was N2
A: (Which) (N1) (was) (N2)

Q: Who employed , met , and was influenced by N1 and married M1
A: (Who) ((employed , met , and was influenced by N1) and (married M1))

Q: Were N1 executive produced by , written by , and directed by N2 , and produced by N3
A: (Were) (N1) ((executive produced by , written by , and directed by N2) , and (produced by N3))

Q: What N1 married , divorced , and was kissed by N2
A: (What) (N1) (married , divorced , and was kissed by) (N2)

Q: Which N1 did M0 acquire
A: (Which) (N1) (did) (M0) (acquire)

Q: Which N1 was influenced by N2 , influenced by N3 , and influenced by M4
A: (Which) (N1) ((was influenced by N2) , (influenced by N3) , and (influenced by M4))

Q: Was N1 employed by N2 and employed by M4 , M5 , and M6 N4
A: (Was) ((N1) ((employed by N2) and (employed by N3))) (N4)

Q: Which N1 did N2 produce and M1 direct
A: (Which) (N1) (did) ((N2 produce) and (M1 direct))

Q: Did N1 influence N2 and influence N3
A: (Did) (N1) ((influence N2) and (influence N3))

Q: What N1 did N2 influence
A: (What) (N1) (did) (N2) (influence)

Q: Did M0 direct , produce , executive produce , edit , and write N1
A: (Did) (M0) (direct , produce , executive produce , edit , and write) (N1)

Q: Was M0 executive produced by , edited by , written by , and directed by N1
A: (Was) (M0) (executive produced by , edited by , written by , and directed by) (N1)

Q: Who influenced and was influenced by N1
A: (Who) (influenced and was influenced by) (N1)

Q: Did N1 executive produce and produce M1
A: (Did) (N1) (executive produce and produce) (M1)

Q: Did N1 found N1 and found M2
A: (Did) (N1) ((found N1) and (found M2))

Q: What was executive produced by N1 and executive produced by N2
A: (What) ((was executive produced by N1) and (executive produced by N2))

Q: Was N1 produced by and written by N2
A: (Was) (N1) (produced by and written by) (N2)

Q: What did N1 produce and write

A: (What) (did) (N1) (produce and write)

Q: Was M1 produced by N1
A: (Was) (M1) (produced by) (N1)

Q: Did N1 edit , N2 direct , and N3 produce M4
A: (Did) ((N1 edit) , (N2 direct) , and (N3 produce)) M4

Q: Was N1 written by and executive produced by N2 M1
A: (Was) ((N1) (written by and executive produced by) (N2)) (M1)

Q: Which N1 was founded by N2
A: (Which) (N1) (was founded by) (N2)

Q: Which N1 was acquired by N2 and acquired N3
A: (Which) (N1) ((was acquired by N2) and (acquired N3))

Q: Was M2 N0 written by M4 and directed by N1
A: (Was) (M2) ((N0) ((written by M4) and (directed by N1)))

Q: What N1 did N2 marry and influence
A: (What) (N1) (did) (N2 marry and influence)

Q: Did N1 influence and marry N2
A: (Did) (N1) (influence and marry) (N2)

Q: Which N1 did M1 marry and M2 marry
A: (Which) (N1) (did) ((M1 marry) and (M2 marry))

Q: What was directed by and edited by N1
A: (What) (was directed by and edited by) (N1)

Q: What did N1 edit and M0 executive produce
A: (What) (did) ((N1 edit) and (M0 executive produce))

Q: What N0 did N1 edit and produce
A: (What) (N0) (did) (N1) (edit and produce)

Q: Was N1 N2 edited by M0
A: (Was) (N1) (N2 (edited by) M0)

Q: Did N1 write a film and direct N2
A: (Did) (N1) ((write a film) and (direct N2))

Q: Was N1 produced by M1 and executive produced by M2
A: (Was) (N1) ((produced by M1) and (executive produced by M2))

Q: Were N1 written , executive produced , produced , and edited by N2
A: (Were) (N1) (written , executive produced , produced , and edited by) (N2)

Q: Was N0 influenced by N1 and influenced by M1 M1
A: (Was) ((N0) (influenced by N1) and (influenced by M1)) (M1)

Q: What was edited by M0 and executive produced by N0
A: (What) ((was edited by M0) and (executive produced by N0))

Q: Did M2 star M3 and star N0
A: (Did) (M2) ((star M3) and (star N0))

Q: Was M1 employed by M2 and employed by N0
A: (Was) (M1) ((employed by M2) and (employed by N0))

Q: What did N0 write , direct , edit , executive produce , and produce
A: (What) (did) (N0) (write , direct , edit , executive produce , and produce)

Q: Was N1 N2 founded by N3 and founded by N4
A: (Was) (N1) (N2 ((founded by N3) and (founded by N4)))

Q: Did M2 marry N1 employed by N2
A: (Did) (M2) (influence) (N1 employed by N2)

### A.1.4   STEP 4: PART OF SPEECH TAGGING

**Part of speech tagging of noun phrases**
Q: M1
A: M=(M1)

Q: (film)
A: N=(film)

Q = N2
A: N=(N2)

Q: M0 's producer
A: M=(M0) P=('s) N=(producer)

Q: a art director 's sibling
A: [a] (N=(art director) P=('s) N=(sibling))

Q: a French producer and editor 's child 's spouse
A: [a] ((N=(French /A, producer, [and] editor) P=('s) N=(child)) P=('s) N=(spouse))

Q: female French costume designer , producer , and editor of M3 and M4
A: N=(female /A, French /A, costume designer, producer, [and] editor) P=(of) M=(M3, M4)

Q: country of nationality of M3
A: N=(country of nationality) P=(of) M=(M3)

Q: a Dutch parent and spouse
A: [a] N=(Dutch /A, parent, [and] spouse)

Q: a editor , producer , and writer of a film 's prequel
A: [a] (N=(editor, producer, [and] writer) P=(of) ([a] (N=(film) P=('s) N=(prequel))))

Q: a female film director 's Spanish parent
A: [a] ((female /A, film director) P=('s) N=(Spanish /A, parent))

Q: M3 and M6
A: M=(M3, [and] M6)

Q: a writer 's spouse , friend , and employer
A: [a] (N=(writer) P=('s) N=(spouse, friend, [and] employer))

Q: male Spanish spouse and parent
A: N=(male /A, Spanish /A, spouse, [and] parent)

Q: a person
A: [a] N=(person)

Q: M4 , M5 , M6 , and M7
A: M=(M4, M5, M6, [and] M7)

Q: film director and editor of M0 , M1 , and M2
A: N=(film director, [and] editor) P=(of) M=(M0, M1, [and] M2)

Q: a Spanish editor and producer of M5
A: [a] (N=(Spanish /A, editor, [and] producer) P=(of) M=(M5))

Q: child of a production company
A: N=(child) P=(of) ([a] N=(production company))

Q: M1 's sibling and spouse
A: M=(M1) P=('s) N=(sibling, [and] spouse)

Q: M3 's employer 's Mexican employee
A: (M=(M3) P=('s) N=(employer)) P=('s) N=(Mexican /A, employee)

Q: M3 's parent 's employee
A: (M=(M3) P=('s) N=(parent)) P=('s) N=(employee)

Q: M2 's founder and employee
A: M=(M2) P=('s) N=(founder, [and] employee)

Q: director of M3 , M4 , and M5
A: N=(director) P=(of) M=(M3 , M4 , [and] M5)

Q: male director of M3 and M4
A: N=(male /A, director) P=(of) M=(M3, [and] M4)

Q: a child of M4 's male editor 's sibling
A: [a] (N=(child) P=(of) ((M=(M4) P=('s) N=(male /A, editor)) P=('s) N=(sibling)))

Q: spouse of M0 's sibling 's employee
A: N=(spouse) P=(of) ((M=(M0) P=('s) N=(sibling)) P=('s) N=(employee))

Q: a actor 's country of nationality
A: [a] (N=(actor) P=('s) N=(country of nationality))

Q: a female person 's parent 's sibling
A: [a] ((N=(female /A, person) P=('s) N=(parent)) P=('s) N=(sibling))

Q: a screenwriter 's French parent
A: [a] (N=(screenwriter) P=('s) N=(French /A, parent))

Q: sibling and friend of a director and actor
A: N=(sibling, [and] friend) P=(of) ([a] (N=(director, [and] actor)))

Q: Mexican American parent
A: N=(Mexican /A, American /A, parent)

**Part of speech tagging of other phrases**
Q: What
A: W=(What)

Q: Which
A: W=(Which)

Q: Who
A: W=(Who)

Q: (Did)
A: V=(Did)

Q: Was
A: V=(Was)

Q: employed , met , and was influenced by N1
A: V=(employed, met, [and] [was] influenced by) (N1)

Q: married M1
A: V=(married) M=(M1)

Q: executive produced by , written by , and directed by N1
A: V=(executive produced by, V=written by, [and] V=directed by) (N1)

Q: produced by N1
A: V=(produced by) (N1)

Q: M0 was employed by
A: M=(M0) V=([was] employed by)

Q: N1 married
A: (N1) V=(married)

Q: M4 , M5 , M6 , and M7
A: M=(M4, M5, M6, [and] M7)

Q: was written by M1
A: V=([was] written by) M=(M1)

Q: married , divorced , and was kissed by
A: V=(married, divorced, [and] [was] kissed by)

Q: N1 wrote M3
A: (N1) V=(wrote) M=(M3)

Q: was influenced by N1
A: V=([was] influenced by) (N1)

Q: influenced by N1
A: V=(influenced by) (N1)

Q: (employed by M1 , M2 , and M3)
A: V=(employed by) M=(M1, M2, [and] M3)

Q: N1 produce
A: (N1) V=(produce)

Q: influence N1
A: V=(influence) (N1)

Q: M1 employed and was founded by
A: M=(M1) V=(employed, [and] [was] founded by))

Q: direct , produce , executive produce , edit , and write
A: V=(direct, produce, executive produce, edit, [and] write)

Q: executive produced by , edited by , written by , and directed by
A: V=(executive produced by, edited by, written by, [and] directed by)

### A.1.5 STEP 5: VERB NORMALIZATION

Q: and
A: and

Q: was
A: be

Q: Was
A: be

Q: were
A: be

Q: Were
A: be

Q: did
A: do

Q: Did
A: do

Q: directed
A: direct

Q: edited
A: edit

Q: met
A: meet

Q: found
A: found

Q: edited by
A: edit by

Q: written by
A: write by

## A.2 CFQ EXEMPLAR SELECTION: DETAILS

For CFQ, we use a random subset consisting of 1000 examples from the training set as a pool of candidate exemplars. For each input question to process, we then dynamically select exemplars from this pool such that they collectively demonstrate relevant knowledge needed to translate the input sentences. This is done by making sure that as many nodes as possible of the decomposition tree of the input are covered by the decomposition trees of the selected exemplars. Depending on the complexity of the input and the similarity of the candidate exemplars in the pool, we select between 4 and 35 exemplars for any given input.

We provide a general description of this process in Section 3.2 and add the CFQ-specific details here. In Section A.3, we also show the set of all selected exemplars for the example in Figure 6.

### A.2.1 TOP-DOWN MATCHING

We want to select exemplars that cover the structure of the input question as well as possible. We do this using the following process:

We first convert the decomposition trees of all the candidate exemplars as well as the concrete input question into syntactic templates by anonymizing concrete leaves and just keeping their types. For instance, the question shown in Figure 6 results in the syntactic template "Which N (V (N that (V (M , M , [and] M))) and (V (M 's N))".

Then we try to find exemplars that match the full template of the input question. If we succeed, we keep them. Otherwise, we reduce the templates by collapsing some of the nodes. For example, we can collapse the node "(M , M , [and] M)" in the above template and instead just use "M". We again try to find exemplars that match the reduced template, keep them if we succeed, and otherwise continue reducing the templates. We do this until we retrieve exemplars that collectively cover the input template as well as possible.

We wrote a small amount of Python code that implements a generic version of these heuristics and use it for both CFQ and COGS. For the example shown in Figure 6, the top-down matching yields exemplars such as the ones shown below. Note that to provide the LM with additional hints, we add to the original question parentheses that indicate the syntactic structure.

Q: Which cinematographer founded (M1 's distributor)
A: SELECT DISTINCT ?x0 WHERE { ?x0 a cinematographer . ?x0 founded ?x1 . ?x1 distributed M1 }

Q: What (film director ((influenced by M1) and (influenced by (a person that (founded (M4 and M5)))))) produced M2
A: SELECT DISTINCT ?x0 WHERE { ?x0 a film_director . ?x0 produced M2 . ?x0 influenced_by ?x1 . ?x0 influenced_by M1 . ?x1 a person . ?x1 founded M4 . ?x1 founded M5 }

### A.2.2 BOTTOM-UP MATCHING

We also want to select exemplars that collectively cover each of the unanonymized leaves. In our running example, this means that we want to cover the leaves "film editor", "influence", "cinematographer", "write", and "editor". In addition, we prefer exemplars where these leaves occur within a similar syntactic template as in the input question.

For each leaf, we do this by converting the decomposition trees into a form where everything but this leaf is anonymized. For the leaf "editor", this results in "Which N (V (N that (V (M , M , [and] M))) and (V (M 's editor))". Then we try to find exemplars that share as many subtrees containing "editor" as possible. This yields exemplars such as:

Q: Was a (Dutch film editor) M0
A: SELECT count(*) WHERE { M0 a film_editor . M0 has_nationality Dutch }

Q: Which actor (was influenced by and influenced) (M1 's editor)
A: SELECT DISTINCT ?x0 WHERE { ?x0 a actor . ?x0 influenced ?x1 . ?x0 influenced_by ?x1 . ?x1 edited M1 }

Q: Was a (Chinese cinematographer) M0
A: SELECT count(*) WHERE { M0 a cinematographer . M0 has_nationality Chinese }

Q: What did (M0 's sibling) write
A: SELECT DISTINCT ?x0 WHERE { ?x0 written_by ?x1 . ?x1 sibling_of M0 }

Q: Was M1 ((founded by (M0 's editor)) and (founded by M2))
A: SELECT count(*) WHERE { ?x0 edited M0 . M1 founded_by ?x0 . M1 founded_by M2 }

### A.3 CFQ Solution: Details and Prompt

As we discussed in Section 3.3, one novelty of dynamic least-to-most prompting is that we cannot translate the constituents in isolation because they might not correspond to well-formed questions and their translation may depend on the context. Instead, we linearize the composition tree into a sequence of increasingly complex subquestions.

This linearization is performed by a walk over the parse tree. We keep the most generic variant of each top-level node and then expand these nodes step by step to obtain a linear sequence of well-formed subquestions. For each subquestion, we then query the language model using a prompt that consists of three parts.

1. Static prompt context for question grounding
2. Dynamically selected exemplars as additional context
3. Sequential questions

These parts are detailed throughout the rest of this section.

#### A.3.1 Part 1: Static prompt context.

Because we cannot translate constituents in isolation, even the simplest subquestion may be too complex to be translated correctly based on the selected exemplars alone. This is especially the case if the exemplar pool does not contain similar questions.

To teach the language model how to translate the simplest subquestions, we therefore provide it with a constant prompt context consisting of 12 grounding examples that illustrate these kinds of subquestions. In addition to the question and its translation, each of these grounding examples also provides a rationale that tells the model how the translation can be obtained (this resembles our chain-of-thought prompt).

The static prompt context is provided below. Note that we we use a slightly different prefix ("Partial Q: " instead of "Q:") for these grounding questions. This allows us to encourage the model to perform rationale-based reasoning when asking it to translate the simplest question of a sequence. Also note that we again use parentheses to indicate the syntactic structure of the questions.

Partial Q: Was a (costume designer 's parent) (M0 's editor)
Rational: Was = {}, "costume designer 's parent" = { ?x0 parent_of ?x1 . ?x1 a costume_designer }, "M0 's editor" = { ?x0 edited M0 } ==> A: SELECT count(*) WHERE { ?x0 parent_of ?x1 . ?x1 a costume_designer . ?x0 edited M0 }

Partial Q: Was M0 a (screenwriter 's spouse)
Rationale: Was = {}, "M0" replaces ?x0, "screenwriter 's spouse" = { ?x0 married_to ?x1 . ?x1 a writer } ==> A: SELECT count(*) WHERE { M0 married_to ?x1 . ?x1 a writer }

Partial Q: Was a (sequel of M1) M0
Rationale: Was = {}, "star of M1" = { ?x0 has_prequel M1 }, "M0" replaces ?x0 ==> A: SELECT count(*) WHERE { M0 has_prequel M1 }

Partial Q: Was M1 executive produced by a (sibling of a film producer)
Rationale: Was = {}, "M1" replaces ?x0, "executive produced by" = { ?x0 executive_produced_by ?x1 }, "sibling of a film producer" = { ?x1 sibling_of ?x2 . ?x2 a film_producer } ==> A: SELECT count(*) WHERE { M1 executive_produced_by ?x1 . ?x1 sibling_of ?x2 . ?x2 a film_producer }

Partial Q: Did a (film 's prequel) star M1
Rationale: Did = {}, "film 's prequel" = { ?x0 has_sequel ?x1 . ?x1 a film }, "star M1" = { ?x0 starred M1 } ==> A: SELECT count(*) WHERE { ?x0 has_sequel ?x1 . ?x1 a film . ?x0 starred M1 }

Partial Q: Did M0 art direct M1
Rationale: Did = {}, "M0" replaces ?x0, art direct M1 = { ?x0 art_directed M1 } ==> A: SELECT count(*) WHERE { M0 art_directed M1 }

Partial Q: Which person did M1 star
Rationale: Which = {}, "person" = { ?x0 a person }, "did M1 star" = { M1 starred ?x0 } ==> A: SELECT DISTINCT ?x0 WHERE { ?x0 a person . M1 starred ?x0 }

Partial Q: Which (parent of M0) was a (person 's parent)
Rationale: Which = {}, "parent of M0" = { ?x0 parent_of M0 }, "costume designer 's parent" = { ?x0 parent_of ?x1 . ?x1 a costume_designer } ==> A: SELECT DISTINCT ?x0 WHERE { ?x0 parent_of M0 . ?x0 parent_of ?x1 . ?x1 a costume_designer }

Partial Q: What was a film written by M1
Rationale: What = {}, "film written by M1" = { ?x0 a film . ?x0 written_by M1 } ==> A: SELECT DISTINCT ?x0 WHERE { ?x0 a film . ?x0 written_by M1 }

Partial Q: What (star of M1) was a (cinematographer 's parent)",
Rationale: What = {}, "star of M1" = { ?x0 starred_in M1 }, "cinematographer 's parent" = { ?x0 parent_of ?x1 . ?x1 a cinematographer } ==> A: SELECT DISTINCT ?x0 WHERE { ?x0 starred_in M1 . ?x0 parent_of ?x1 . ?x1 a cinematographer }

Partial Q: Who was a (producer of M1)
Rationale: "Who" = { ?x0 a person }, "executive producer of M1" = { ?x0 produced M1 } ==> SELECT DISTINCT ?x0 WHERE { ?x0 a person . ?x0 produced M1 }

Partial Q: Who employed a person influenced by M0
Rationale: "Who" = { ?x0 a person }, "employed a person" = { ?x0 employed ?x1 . ?x1 a person }, "influenced by M0" = { ?x1 influenced_by M0 } ==> A: SELECT DISTINCT ?x0 WHERE { ?x0 a person . ?x0 employed ?x1 . ?x1 a person . ?x1 influenced_by M0 }

### A.3.2 PART 2: DYNAMICALLY SELECTED EXEMPLARS

After the constant prompt context, we add as additional context the exemplars that we dynamically selected for the given input using the process described in Section A.2.

For our example from Figure 6, this results in the following prompt context, which is appended to the static context shown above.

Q: Were (M2 and M3) produced by (a actor that (influenced (M1 's editor)))
A: SELECT count(*) WHERE { ?x0 a actor . ?x0 influenced ?x1 . ?x1 edited M1 . M2 produced_by ?x0 . M3 produced_by ?x0 }

Q: Did M1 ((star a (editor of M0)) and (star M2))
A: SELECT count(*) WHERE { ?x0 edited M0 . M1 starred ?x0 . M1 starred M2 }

Q: Was M1 ((founded by (M0 's editor)) and (founded by M2))
A: SELECT count(*) WHERE { ?x0 edited M0 . M1 founded_by ?x0 . M1 founded_by M2 }

Q: Which actor ((was influenced by (M4 's (art director and writer))) , (influenced M1) , and (influenced (M2 and M3)))
A: SELECT DISTINCT ?x0 WHERE { ?x0 a actor . ?x0 influenced M1 . ?x0 influenced M2 . ?x0 influenced M3 . ?x0 influenced_by ?x1 . ?x1 art_directed M4 . ?x1 wrote M4 }

Q: Which actor ((married a film editor) and (was influenced by (M1 's sibling)))
A: SELECT DISTINCT ?x0 WHERE { ?x0 a actor . ?x0 influenced_by ?x1 . ?x0 married_to ?x2 . ?x1 sibling_of M1 . ?x2 a film_editor }

Q: Did a (film editor 's parent) write M0
A: SELECT count(*) WHERE { ?x0 wrote M0 . ?x0 parent_of ?x1 . ?x1 a film_editor }

Q: Did M0 (write , edit , and produce) M1
A: SELECT count(*) WHERE { M0 edited M1 . M0 produced M1 . M0 wrote M1 }

Q: What did (M0 's sibling) write
A: SELECT DISTINCT ?x0 WHERE { ?x0 written_by ?x1 . ?x1 sibling_of M0 }

Q: Was a (Chinese cinematographer) M0
A: SELECT count(*) WHERE { M0 a cinematographer . M0 has_nationality Chinese }

Q: Who was (a cinematographer that (distributed M2))
A: SELECT DISTINCT ?x0 WHERE { ?x0 a cinematographer . ?x0 a person . ?x0 distributed M2 }

Q: Who ((was influenced by M1) and (influenced by a (Swedish cinematographer)))
A: SELECT DISTINCT ?x0 WHERE { ?x0 a person . ?x0 influenced_by ?x1 . ?x0 influenced_by M1 . ?x1 a cinematographer . ?x1 has_nationality Swedish }

Q: Who ((was influenced by (M1 's star)) and (married (M2 's sibling)))
A: SELECT DISTINCT ?x0 WHERE { ?x0 a person . ?x0 influenced_by ?x1 . ?x0 married_to ?x2 . ?x1 starred_in M1 . ?x2 sibling_of M2 }

Q: Who ((was influenced by M1) and (influenced by (a person that (edited M2))))
A: SELECT DISTINCT ?x0 WHERE { ?x0 a person . ?x0 influenced_by ?x1 . ?x0 influenced_by M1 . ?x1 a person . ?x1 edited M2 }

Q: Which actor (was influenced by and influenced) (M1 's editor)
A: SELECT DISTINCT ?x0 WHERE { ?x0 a actor . ?x0 influenced ?x1 . ?x0 influenced_by ?x1 . ?x1 edited M1 }

Q: Was a (Dutch film editor) M0
A: SELECT count(*) WHERE { M0 a film_editor . M0 has_nationality Dutch }

Q: Was M1 a (employer of a film editor)
A: SELECT count(*) WHERE { ?x0 a film_editor . M1 employed ?x0 }

Q: Did a (actor of M1) edit M0
A: SELECT count(*) WHERE { ?x0 portrayed M1 . ?x0 edited M0 }

Q: Did M1 ((star M2) and (star (M0 's employee)))
A: SELECT count(*) WHERE { ?x0 employed_by M0 . M1 starred ?x0 . M1 starred M2 }

Q: Did M0 (executive produce (M1 , M2 , and produce)) M3
A: SELECT count(*) WHERE { M0 executive_produced M1 . M0 executive_produced M2 . M0 produced M3 }

Q: What (film director ((influenced by M1) and (influenced by (a person that (founded (M4 and M5)))))) produced M2
A: SELECT DISTINCT ?x0 WHERE { ?x0 a film_director . ?x0 produced M2 . ?x0 influenced_by ?x1 . ?x0 influenced_by M1 . ?x1 a person . ?x1 founded M4 . ?x1 founded M5 }

Q: Which cinematographer founded (M1 's distributor)
A: SELECT DISTINCT ?x0 WHERE { ?x0 a cinematographer . ?x0 founded ?x1 . ?x1 distributed M1 }

Q: Which character influenced a film editor
A: SELECT DISTINCT ?x0 WHERE { ?x0 a fictional_character . ?x0 influenced ?x1 . ?x1 a film_editor }

Q: Which film editor founded (M1 's distributor)
A: SELECT DISTINCT ?x0 WHERE { ?x0 a film_editor . ?x0 founded ?x1 . ?x1 distributed M1 }

Q: Which character married a screenwriter
A: SELECT DISTINCT ?x0 WHERE { ?x0 a fictional_character . ?x0 married_to ?x1 . ?x1 a writer }

### A.3.3 PART 3: SEQUENTIAL QUESTIONS

After providing the static and dynamic prompt context, we start by grounding the simplest subquestion. To encourage the model to use the rationale-based reasoning, we use the prefix "Partial Q:" rather than just "Q:" for this question. After we obtain the translation from the model, we append it to the prompt and then again ask the same subquestion, but this time using the regular "Q:" prefix. In addition, we tell the model that it can simply copy the answer from above by adding the comment "# Copy the answer from above".

For the example from Figure 6, this first part of the prompt looks as follows. Note that the text that is not bolded corresponds to the answers provided by the model.

**Partial Q: Which film editor was influenced by a cinematographer**
Rationale: Which = {}, "film editor" = { ?x0 a film_editor }, "was influenced by a cinematographer" = { ?x0 influenced_by ?x1 . ?x1 a cinematographer } ==> A: SELECT DISTINCT ?x0 WHERE { ?x0 a film_editor . ?x0 influenced_by ?x1 . ?x1 a cinematographer }

**# Copy the answer from above.**
**Q: Which film editor was influenced by a cinematographer**
A: SELECT DISTINCT ?x0 WHERE { ?x0 a film_editor . ?x0 influenced_by ?x1 . ?x1 a cinematographer }

After grounding the simplest question, we continue with the next subquestions, which are presented to the model one after the other using the ordinary prefix "Q:". Sometimes, a subquestion is a strict extension of the previous subquestion. To make sure that the model does not miss this, we add a comment of the form "# Extend the answer above. Add: ...". which highlights the delta between the questions. In other cases, a subquestion is almost the same as the previous question, except that one constituent is replaced by another.

For each of these subquestions, we perform a request to the language model and append the answer to the prompt before asking the next question. For our running example, this part of the prompt looks as follows. Note that we obtain the correct answer to the original question at the end.

**# Extend the answer above. Add: that (wrote M3)**
**Q: Which film editor (was influenced by (a cinematographer that (wrote M3)))**
A: SELECT DISTINCT ?x0 WHERE { ?x0 a film_editor . ?x0 influenced_by ?x1 . ?x1 a cinematographer . ?x1 wrote M3 }

**# Extend the answer above. Add: M5**
**Q: Which film editor (was influenced by (a cinematographer that (wrote (M3 and M5))))**
A: SELECT DISTINCT ?x0 WHERE { ?x0 a film_editor . ?x0 influenced_by ?x1 . ?x1 a cinematographer . ?x1 wrote M3 . ?x1 wrote M5 }

**# Extend the answer above. Add: M4**
**Q: Which film editor (was influenced by (a cinematographer that (wrote (M3 , M4 , and M5))))**
A: SELECT DISTINCT ?x0 WHERE { ?x0 a film_editor . ?x0 influenced_by ?x1 . ?x1 a cinematographer . ?x1 wrote M3 . ?x1 wrote M4 . ?x1 wrote M5 }

**Q: Which film editor (influenced by (M1 's editor))**
A: SELECT DISTINCT ?x0 WHERE { ?x0 a film_editor . ?x0 influenced_by ?x1 . ?x1 edited M1 }

**# Extend the answer above. Add: was influenced by (a cinematographer that (wrote (M3 , M4 , and M5)))**
**Q: Which film editor ((was influenced by (a cinematographer that (wrote (M3 , M4 , and M5)))) and (influenced by (M1 's editor)))**
A: SELECT DISTINCT ?x0 WHERE { ?x0 a film_editor . ?x0 influenced_by ?x1 . ?x1 edited M1 . ?x0 influenced_by ?x2 . ?x2 a cinematographer . ?x2 wrote M3 . ?x2 wrote M4 . ?x2 wrote M5 }

## B  LEAST-TO-MOST PROMPTING FOR COGS

**James said that a manager liked that Aiden appreciated that Emily believed that the girl was posted a cake beside a table by Olivia .**
```
PARSE: say ( agent = James , ccomp = like
    ( agent = manager , ccomp = appreciate
    ( agent = Aiden , ccomp = believe
    ( agent = Emily , ccomp = post
    ( recipient = * girl , theme = cake
    ( nmod .  beside = table ) , agent = Olivia ) ) ) ) ) DONE
```

**The boy shortened the donut beside the bed in the car in the garden in the can on the tree .**
```
PARSE: shorten ( agent = * boy , theme = * donut
    ( nmod .  beside = * bed
    ( nmod .  in = * car
    ( nmod .  in = * garden
    ( nmod .  in = * can
    ( nmod .  on = * tree ) ) ) ) ) ) DONE
```

Figure 7: Two COGS example, which we use to illustrate the application of dynamic least-to-most prompting to COGS. The first example contains nested subclauses while the second example contains nested propositional phrases.

In this section, we provide more details on the application of dynamic least-to-most prompting for COGS. In particular, we detail all the prompts and show how the example in Figure 7 is processed step by step. Note that this is a variation of the application of dynamic-least-to-most prompting for CFQ, which is detailed in Section A. Therefore, we mostly focus on highlighting the differences.

### B.1  COGS DECOMPOSITION: DETAILS AND PROMPTS

As discussed in Section 3.1, we use prompting-based syntactic parsing to decompose COGS sentences. To teach LMs to perform this kind of decomposition, we divide the syntactic parsing task into the following steps

1. *Iterative* subclause decomposition
2. Phrase identification
3. *Iterative* prepositional phrase decomposition and noun phrase annotation
4. Verb phrase normalization

This is quite similar to the steps used for CFQ (see Section A.1), but there are some important differences. Since the CFQ dataset only contains limited nesting of subclauses and noun phrases, we were able to identify them in a single step. As exemplified by the first example in Figure 7, the COGS dataset contains much deeper nesting of subclauses (up to level 12), which we address by performing subclause decomposition iteratively. This means that the prompt for subclause decomposition (step 1) is designed such that it only extracts one subclause at a time and is applied iteratively until all subclauses are extracted.

As exemplified by the second example in Figure 7, COGS also contains deep nesting of prepositional phrases (up to level 12). We again address this by performing prepositional phrase decomposition (step 3) iteratively and designed the prompt such that it only extracts one prepositional phrase at a time and is applied iteratively until all prepositional phrases are extracted.

**Example 1.**  To illustrate this step-by-step process, we begin with the first example from Figure 7, which is "James said that a manager liked that Aiden appreciated that Emily believed that the girl was posted a cake beside a table by Olivia .". The first step decomposes the subclauses of this sentence via 4 iterative calls:

- P=(James) V=(said) that C=(a manager liked that Aiden appreciated that Emily believed that the girl was posted a cake beside a table by Olivia)

- P=(a manager) V=(liked) that C=(Aiden appreciated that Emily believed that the girl was posted a cake beside a table by Olivia)
- P=(Aiden) V=(appreciated) that C=(Emily believed that the girl was posted a cake beside a table by Olivia)
- P=(Emily) V=(believed) that C=(the girl was posted a cake beside a table by Olivia)

In the second step, we identify the different phrases for the final subclause "the girl was posted a cake beside a table by Olivia", which yields:

- P=(the girl) V=(was posted) P=(a cake beside a table) by P=(Olivia)

In the third step, we decompose the propositional phrase, which is not nested and therefore requires only one iteration:

- (a cake) beside P=(a table)

Note that the same prompt is also used to annotate noun phrases by marking the use of the definite article with a star. We therefore also apply it to all other noun phrases, which yields:

- James
- a manager
- Aiden
- Emily
- the * girl
- Olivia
- a table

In the fourth step, we normalize all the verbs, which means that "said" yields "say", "liked" yields "like", "appreciated" yields "appreciate", "believed" yields "believe", and "was posted" yields "post". Finally, we run a few lines of Python code that puts all these parts back together to obtain a parse tree. Note that the normalized verbs do not replace the original ones but are kept in addition, which allows us to obtain higher recall when selecting exemplars.

This yields the following fully decomposed sentence:

- (James) (said [say]) that ((a manager) (liked [like]) that ((Aiden) (appreciated [appreciate]) that ((Emily) (believed [believe]) that ((the * girl) (was posted [post]) ((a cake) (beside) (a table)) by (Olivia)))))

**Example 2.** We now walk through the step-by-step process using the second example from Figure 7, which is "The boy shortened the donut beside the bed in the car in the garden in the can on the tree .". Since this example does not contain any suubclauses, the first step does not apply. In the second step, we identify the different phrases, which yields:

- P=(The boy) V=(shortened) P=(the donut beside the bed in the car in the garden in the can on the tree)

In the third step, we decompose the prepositional phrase, which happens to be nested and therefore requires 5 iterations:

- (the * donut) beside P=(the bed in the car in the garden in the can on the tree)
- (the * bed) in P=(the car in the garden in the can on the tree)
- (the * car) in P=(the garden in the can on the tree)
- (the * garden) in P=(the can on the tree)
- (the * can) on P=(the tree)

Since the same prompt is also used to annotate noun phrases by marking the use of the definite article with a star, we also apply it to all other noun phrases, which yields:

- The * boy
- the * tree

In the fourth step, we normalize all the verbs, which means that "shortened" yields "shorten".

This yields the following fully decomposed sentence:

- (The * boy) (shortened [shorten]) ((the * donut) (beside) ((the * bed) (in) ((the * car) (in) ((the * garden) (in) ((the * can) (on) (the * tree))))))

Below, we present the exact prompt contexts used for each of the parsing steps.

### B.1.1 ITERATIVE SUBCLAUSE DECOMPOSITION

Q: The girl expected that Daniel liked that a weapon was liked
A: P=(The girl) V=(expected) that C=(Daniel liked that a weapon was liked)

Q: The girl hoped that a donut was returned to a cat by a monkey
A: P=(The girl) V=(hoped) that C=(a donut was returned to a cat by a monkey)

Q: a teacher thought that a girl liked that a cookie was valued by Emma
A: P=(a teacher) V=(thought) that C=(a girl liked that a cookie was valued by Emma)

Q: Benjamin supported that Avery loaned a donut in a cup to a frog
A: P=(Benjamin) V=(supported) C=(that Avery loaned a donut in a cup to a frog)

Q: A girl liked that a frog liked that Eleanor gave a brush to the professor
A: P=(A girl) V=(liked) that C=(a frog liked that Eleanor gave a brush to the professor)

Q: Charlotte expected that Emma liked that the patient lended a box on a table to Nora
A: P=(Charlotte) V=(expected) that C=(Emma liked that the patient lended a box on a table to Nora)

Q: Aiden believed that a mouse hoped that Olivia examined a glue beside the stage beside a duck
A: P=(Aiden) V=(believed) that C=(a mouse hoped that Olivia examined a glue beside the stage beside a duck)

Q: a girl declared that the boy was offered the watch beside the table
A: P=(a girl) V=(declared) that C=(the boy was offered the watch beside the table)

Q: the teacher declared that a donut was given to the student by Noah
A: P=(the teacher) V=(declared) that C=(a donut was given to the student by Noah)

Q: A dog respected that Emma rented the biscuit in a pot beside a nest to the boy
A: P=(A dog) V=(respected) that C=(Emma rented the biscuit in a pot beside a nest to the boy)

Q: The teacher liked that a driver hoped that the girl meant to eat
A: P=(The teacher) V=(liked) that C=(a driver hoped that the girl meant to eat)

Q: the giraffe liked that Olivia noticed that the butterfly was given the drink in the garden on a table
A: P=(the giraffe) V=(liked) that C=(Olivia noticed that the butterfly was given the drink in the garden on a table)

Q: A cat in a box on a table tolerated that a professor beside the stage liked a dog
A: P=(A cat in a box on a table) V=(tolerated) that C=(a professor beside the stage liked a dog)

Q: the professor on a chair beside a bench in a room expected that a mouse was given a present by a doctor on the stage in a house
A: P=(the professor on a chair beside a bench in a room) V=(expected) that C=(a mouse was given a present by a doctor on the stage in a house)

Q: Joe liked that Fred thought that a girl confessed that the chicken meant that Freda liked that Peter hoped that Elizabeth said that a professor whished that Anna hoped that Lisa declared that the creature tolerated that a teacher liked that Lily was given the clock beside a guitar beside the table by Isabella
A: P=(Joe) V=(liked) that C=(Fred thought that a girl confessed that the chicken meant that Freda liked that Peter hoped that Elizabeth said that a professor whished that Anna hoped that Lisa declared that the creature tolerated that a teacher liked that Lily was given the clock beside a guitar beside the table by Isabella)

### B.1.2 PHRASE IDENTIFICATION

Q: a boy meant to talk
A: P=(a boy) V=(meant) (to talk)

Q: Camila rolled a lamb beside a flower
A: P=(Camila) V=(rolled) P=(a lamb beside a flower))

Q: the hen appreciated the journalist on the piano
A: P=(the hen) V=(appreciated) P=(the journalist on the piano)

Q: The chicken on a box needed to eat
A: P=(The chicken on a box) V=(needed) (to eat)

Q: a king awarded Joe Fred
A: P=(a king) V=(awarded) P=(Joe) P=(Fred)

Q: The drink was snapped by Ava
A: P=(The drink) V=(was snapped) by P=(Ava)

Q: the teacher passed Joe a pen
A: P=(the teacher) V=(passed) P=(Joe) P=(a pen)

Q: the girl offered Fred to James
A: P=(the girl) V=(offered) P=(Fred) to P=(James)

Q: A teacher beside the table wanted to hunt
A: P=(A teacher beside the table) V=(wanted) (to hunt)

Q: A crayon was handed to the girl on the table
A: P=(A crayon) V=(was handed) to P=(the girl on the table)

Q: Aria rented the boy a donut in the room
A: P=(Aria) V=(rented) P=(the boy) P=(a donut in the room)

Q: Peter mailed Sawyer the hedgehog in the garage
A: P=(Peter) V=(mailed) P=(Sawyer) P=(the hedgehog in the garage)

Q: a boy froze the sandwich on the stage in the room
A: P=(a boy) V=(froze) P=(the sandwich on the stage in the room)

Q: A donut was returned to a cat beside the dog by a monkey on a roof
A: P=(A donut) V=(was returned) to P=(a cat beside the dog) by P=(a monkey on a roof)

Q: Avery loaned a donut in a cup to a frog
A: P=(Avery) V=(loaned) P=(a donut in a cup) to P=(a frog)

Q: the patient lended a box on a table to Nora
A: P=(the patient) V=(lended) P=(a box on a table) to P=(Nora)

Q: The butterfly was given the drink in the garden on a table
A: P=(The butterfly) V=(was given) P=(the drink in the garden on a table)

Q: Luke gave the cake on the windowsill beside a bed under a lamp to a teacher on a pony
A: P=(Luke) V=(gave) P=(the cake on the windowsill beside a bed under a lamp) to P=(a teacher on a pony)

Q: The teacher received a cat in a box on a table on a carpet beside a window from Joe
A: P=(The teacher) V=(received) P=(a cat in a box on a table on a carpet beside a window) (from P=(Joe))

Q: A cat in a cage on a table gave a present in a box to a dog on the floor
A: P=(A cat in a cage on a table) V=(gave) P=(a present in a box) to P=(a dog on the floor)

Q: A ring on a pillow in a wrapper was presented to the professor in a room by a priest on a chair
A: P=(a ring on a pillow in a wrapper) V=(was presented) to P=(the professor in a room) by P=(a priest on a chair)

Q: Charles posted the pony in a barn the drink in a pear on the floor beside the book in the iron on a brick in the robe in a pipe
A: P=(Charles) V=(posted) P=(the pony in a barn) P=(the drink in a pear on the floor beside the book in the iron on a brick in the robe in a pipe)

Q: a frog was given the apple in a school in a town on a stool on a sheet on a table on a desk on the floor in a room beside a building on a road in a city in the world in the universe by a baby
A: P=(a frog) V=(was given) P=(the apple in a school in a town on a stool on a sheet on a table on a desk on the floor in a room beside a building on a road in a city in the world in the universe) by P=(a baby)

### B.1.3 Iterative prepositional phrase decomposition and noun phrase annotation

Q: Joe
A: Joe

Q: Liam
A: Liam

Q: Isabelle
A: Isabelle

Q: Benedict
A: Benedict

Q: A atom
A: A atom

Q: a helmet
A: a helmet

Q: A camera
A: A camera

Q: a helmet
A: a helmet

Q: the atom
A: the * atom

Q: The helmet
A: The * helmet

Q: the camera
A: the * camera

Q: The song
A: The * song

Q: the rino
A: the * rino

Q: The lizzard
A: The * lizzard

Q: the plane
A: the * plane

Q: The gnome
A: The * gnome

Q: the nose
A: the * nose

Q: The adventure
A: The * adventure

Q: the undertaking
A: the * undertaking

Q: The camera
A: The * camera

Q: the sky
A: the * sky

Q: The trainer
A: The * trainer

Q: the underdog
A: the * underdog

Q: The mole
A: The * mole

Q: the floor
A: the * floor

Q: The stone
A: The * stone

Q: the smartphone

A: the * smartphone

Q: The hiccups
A: The * hiccups

Q: the wine
A: the * wine

Q: The dinosaur
A: The * dinosaur

Q: the boy
A: the * boy

Q: a atom beside a flower
A: (a atom) (beside) P=(a flower)

Q: A song beside a roof
A: (A song) (beside) P=(a roof)

Q: the shark in the helmet
A: (the * shark) (in) P=(the helmet)

Q: The camera beside the pool
A: (The * camera) (beside) P=(the pool)

Q: the floor in the rino
A: (the * floor) (in) P=(the rino)

Q: the sky on a ceiling
A: (the * sky) (on) P=(a ceiling)

Q: the plane on the piano
A: (the * plane) (on) P=(the piano)

Q: A rino in a trainer
A: (A rino) (in) P=(a trainer)

Q: The sandwich on the underdog in the camera
A: (The * sandwich) (on) P=(the underdog in the camera)

Q: A undertaking in the camera on a leg
A: (A undertaking) (in) P=(the camera on a leg)

Q: The helmet on the smartphone
A: (The * helmet) (on) P=(the smartphone)

Q: the helmet on the dinosaur beside a smartphone under a wine
A: (the * helmet) (on) P=(the dinosaur beside a smartphone under a wine)

Q: A stone in a trainer on a leg on a helmet beside a window
A: (A stone) (in) P=(a trainer on a leg on a helmet beside a window)

Q: a riffle on the floor in the rino on the leg beside the stone on a spear beside the rino on a sofa
A: (a riffle) (on) P=(the floor in the rino on the leg beside the stone on a spear beside the rino on a sofa)

Q: The referee in a rino beside a official on a smartphone in the camera beside the trainer in

a trainer under the spider on the shingle
A: (The * referee) (in) P=(a rino beside a official on a smartphone in the camera beside the trainer in a trainer under the spider on the shingle)

Q: Joe in a helmet in a vessel on the water beside a harbor in a village beside city in a country on a continent on earth
A: (Joe) (in) P=(a helmet in a vessel on the water beside a harbor in a village beside city in a country on a continent on earth)

Q: a underdog beside a foreigner on a smartphone in a truck in a microwave beside a rocket on a computer on a stool on the surface
A: (a underdog) (beside) P=(a foreigner on a smartphone in a truck in a microwave beside a rocket on a computer on a stool on the surface)

Q: the rooster over a computer on a board on the spear on a leg on the floor beside a rocker under a window on a shingle beside a rino under the clouds on the sky
A: (the * rooster) (over) P=(a computer on a board on the spear on a leg on the floor beside a rocker under a window on a shingle beside a rino under the clouds on the sky)

Q: A phone on the helmet in the rino on the wallet beside the knive on a floor beside the lizzard on a underdog in a wardrobe beside the rocket in the bus beside a hut in the village
A: (A phone) (on) P=(the helmet in the rino on the wallet beside the knive on a floor beside the lizzard on a underdog in a wardrobe beside the rocket in the bus beside a hut in the village)

Q: the country in a school in a town on a stool on a sheet on a leg on a rocker on the floor in a camera beside a building on a street in a city in the world in the universe
A: (the * country) (in) P=(a school in a town on a stool on a sheet on a leg on a rocker on the floor in a camera beside a building on a street in a city in the world in the universe)

### B.1.4 Verb phrase normalization

Q: expected
A: expect

Q: hoped
A: hope

Q: thought
A: think

Q: supported
A: support

Q: liked
A: like

Q: gave
A: give

Q: ate
A: eat

Q: was snapped
A: snap

Q: was given
A: give

Q: was returned

A: return

Q: was lended
A: lend

Q: was brought
A: bring

## B.2 COGS EXEMPLAR SELECTION: DETAILS

We provide a general description of the exemplar selection process in Section 3.2 and add the CFQ-specific details in Section A.2. Since COGS is using the same heuristics with slightly different parameters, we just highlight the differences here.

### B.2.1 EXEMPLAR POOL

For CFQ, we use a random subset consisting of 1000 examples from the training set as a pool of candidate exemplars. For COGS, we went a different route and manually selected a set of exemplars from the training data. The main reason for this is that COGS contains many different verbs and since unergative and unaccusative are translated differently, it is important to include as many of them in the exemplar pool as possible, as it would otherwise be hard for the model to figure out how a certain verb should be treated. Also, since some of those verbs are quite rare in the training data, some of them would not be included in a reasonably sized random sample.

Concretely, we include in the exemplar pool for COGS the following 62 exemplars containing all unergative and unaccusative verbs that occur in the training data. As for CFQ, we use for all exemplars a parsed version of the natural language sentence, which contains additional parentheses and other annotations that make it easier for the model to perform the translation.

(The * goose) (baked [bake])
PARSE: bake ( agent = * goose ) DONE

(A bell) (broke [break])
PARSE: break ( theme = bell ) DONE

(A molecule) (burned [burn])
PARSE: burn ( theme = molecule ) DONE

(A cat) (called [call])
PARSE: call ( agent = cat ) DONE

(A cake) (changed [change])
PARSE: change ( theme = cake ) DONE

(A student) (cleaned [clean])
PARSE: clean ( agent = student ) DONE

(A pickle) (collapsed [collapse])
PARSE: collapse ( theme = pickle ) DONE

(A baby) (cooked [cook])
PARSE: cook ( agent = baby ) DONE

(A dog) (crumpled [crumple])
PARSE: crumple ( theme = dog ) DONE

(A pig) (cried [cry])
PARSE: cry ( agent = pig ) DONE

(A chicken) (danced [dance])

PARSE: dance ( agent = chicken ) DONE

(A buyer) (decomposed [decompose])
PARSE: decompose ( theme = buyer ) DONE

(A mandarin) (disintegrated [disintegrate])
PARSE: disintegrate ( theme = mandarin ) DONE

(A bowl) (doubled [double])
PARSE: double ( theme = bowl ) DONE

(A cat) (drew [draw])
PARSE: draw ( agent = cat ) DONE

(A monster) (dusted [dust])
PARSE: dust ( agent = monster ) DONE

(A boy) (ate [eat])
PARSE: eat ( agent = boy ) DONE

(A boy) (enlarged [enlarge])
PARSE: enlarge ( theme = boy ) DONE

(A boy) (examined [examine])
PARSE: examine ( agent = boy ) DONE

(A girl) (floated [float])
PARSE: float ( theme = girl ) DONE

(A melon) (froze [freeze])
PARSE: freeze ( theme = melon ) DONE

(A resident) (frowned [frown])
PARSE: frown ( agent = resident ) DONE

(A dog) (gasped [gasp])
PARSE: gasp ( agent = dog ) DONE

(A chicken) (giggled [giggle])
PARSE: giggle ( agent = chicken ) DONE

(A dog) (grew [grow])
PARSE: grow ( theme = dog ) DONE

(A butterfly) (heard [hear])
PARSE: hear ( agent = butterfly ) DONE

(A child) (hunted [hunt])
PARSE: hunt ( agent = child ) DONE

(A cat) (improved [improve])
PARSE: improve ( theme = cat ) DONE

(A boy) (inflated [inflate])
PARSE: inflate ( theme = boy ) DONE

(A cat) (investigated [investigate])
PARSE: investigate ( agent = cat ) DONE

(A girl) (jogged [jog])
PARSE: jog ( agent = girl ) DONE

(A chicken) (juggled [juggle])
PARSE: juggle ( agent = chicken ) DONE

(A penguin) (knew [know])
PARSE: know ( agent = penguin ) DONE

(A mouse) (laughed [laugh])
PARSE: laugh ( agent = mouse ) DONE

(A mouse) (napped [nap])
PARSE: nap ( agent = mouse ) DONE

(A professor) (noticed [notice])
PARSE: notice ( agent = professor ) DONE

(A bee) (nursed [nurse])
PARSE: nurse ( agent = bee ) DONE

(A dog) (observed [observe])
PARSE: observe ( agent = dog ) DONE

(A frog) (packed [pack])
PARSE: pack ( agent = frog ) DONE

(A kid) (painted [paint])
PARSE: paint ( agent = kid ) DONE

(A dog) (reddened [redden])
PARSE: redden ( theme = dog ) DONE

(A pig) (rolled [roll])
PARSE: roll ( theme = pig ) DONE

(A queen) (ran [run])
PARSE: run ( agent = queen ) DONE

(A butterfly) (scoffed [scoff])
PARSE: scoff ( agent = butterfly ) DONE

(A dog) (screamed [scream])
PARSE: scream ( agent = dog ) DONE

(A boy) (saw [see])
PARSE: see ( agent = boy ) DONE

(Julian) (shattered [shatter])
PARSE: shatter ( theme = Julian ) DONE

(A cookie) (shortened [shorten])
PARSE: shorten ( theme = cookie ) DONE

(A boy) (sketched [sketch])
PARSE: sketch ( agent = boy ) DONE

(A soldier) (slept [sleep])
PARSE: sleep ( agent = soldier ) DONE

(A raisin) (slid [slide])
PARSE: slide ( theme = raisin ) DONE

(A champion) (smiled [smile])
PARSE: smile ( agent = champion ) DONE

(A chicken) (smirked [smirk])
PARSE: smirk ( agent = chicken ) DONE

(A donut) (snapped [snap])
PARSE: snap ( theme = donut ) DONE

(A tenant) (sneezed [sneeze])
PARSE: sneeze ( agent = tenant ) DONE

(A student) (snoozed [snooze])
PARSE: snooze ( agent = student ) DONE

(A girl) (snored [snore])
PARSE: snore ( agent = girl ) DONE

(A boy) (split [split])
PARSE: split ( theme = boy ) DONE

(A monster) (studied [study])
PARSE: study ( agent = monster ) DONE

(A girl) (stuttered [stutter])
PARSE: stutter ( agent = girl ) DONE

(A crocodile) (talked [talk])
PARSE: talk ( agent = crocodile ) DONE

(A baby) (walked [walk])
PARSE: walk ( agent = baby ) DONE

Once the sentences are fully decomposed, COGS only contains only 9 different types of verb phrases. We select for the pool 3 examples for each of them, which adds another 27 exemplars to the pool (see below) and hence results in a pool size of 89 exemplars in total.

(The * girl) (fed [feed]) (the * dog) (the * mandarin)
PARSE: feed ( agent = * girl , recipient = * dog , theme = * mandarin ) DONE

(The * cat) (gave [give]) (the * host) (the * yogurt)
PARSE: give ( agent = * cat , recipient = * host , theme = * yogurt ) DONE

(The * mouse) (loaned [loan]) (the * turkey) (the * chalk)
PARSE: loan ( agent = * mouse , recipient = * turkey , theme = * chalk ) DONE

(The * creature) (wired [wire]) (the * cake) to (the * giraffe)
PARSE: wire ( agent = * creature , theme = * cake , recipient = * giraffe ) DONE

(The * sailor) (gave [give]) (the * bucket) to (the * chicken)
PARSE: give ( agent = * sailor , theme = * bucket , recipient = * chicken ) DONE

(The * chicken) (brought [bring]) (the * pen) to (the * consumer)
PARSE: bring ( agent = * chicken , theme = * pen , recipient = * consumer ) DONE

(The * cat) (was awarded [award]) (the * pencil)

PARSE: award ( recipient = * cat , theme = * pencil ) DONE

(The * lion) (was given [give]) (the * cake)
PARSE: give ( recipient = * lion , theme = * cake ) DONE

(The * coach) (was lended [lend]) (the * balloon)
PARSE: lend ( recipient = * coach , theme = * balloon ) DONE

(The * frog) (was mailed [mail]) (the * ball) by (the * child)
PARSE: mail ( recipient = * frog , theme = * ball , agent = * child ) DONE

(The * dog) (was given [give]) (the * cookie) by (the * duck)
PARSE: give ( recipient = * dog , theme = * cookie , agent = * duck ) DONE

(The * prince) (was passed [pass]) (the * box) by (the * president)
PARSE: pass ( recipient = * prince , theme = * box , agent = * president ) DONE

(The * doll) (was awarded [award]) to (the * consumer)
PARSE: award ( theme = * doll , recipient = * consumer ) DONE

(The * donut) (was given [give]) to (the * teacher)
PARSE: give ( theme = * donut , recipient = * teacher ) DONE

(The * balloon) (was wired [wire]) to (the * giraffe)
PARSE: wire ( theme = * balloon , recipient = * giraffe ) DONE

(The * bat) (was given [give]) to (the * girl) by (the * governor)
PARSE: give ( theme = * bat , recipient = * girl , agent = * governor ) DONE

(The * cake) (was posted [post]) to (the * hero) by (the * donkey)
PARSE: post ( theme = * cake , recipient = * hero , agent = * donkey ) DONE

(The * block) (was given [give]) to (the * teacher) by (the * butterfly)
PARSE: give ( theme = * block , recipient = * teacher , agent = * butterfly ) DONE

(The * pig) (liked [like]) (the * zebra)
PARSE: like ( agent = * pig , theme = * zebra ) DONE

(The * baby) (missed [miss]) (the * cake)
PARSE: miss ( agent = * baby , theme = * cake ) DONE

(The * creature) (appreciated [appreciate]) (the * present)
PARSE: appreciate ( agent = * creature , theme = * present ) DONE

(The * cake) (was found [find]) by (the * teacher)
PARSE: find ( theme = * cake , agent = * teacher ) DONE

(The * seed) (was eaten [eat]) by (the * coach)
PARSE: eat ( theme = * seed , agent = * coach ) DONE

(The * donut) (was inflated [inflate]) by (Aiden)
PARSE: inflate ( theme = * donut , agent = Aiden ) DONE

(The * shirt) (was liked [like])
PARSE: like ( theme = * shirt ) DONE

(The * rose) (was discovered [discover])
PARSE: discover ( theme = * rose ) DONE

(The * raisin) (was frozen [freeze])
PARSE: freeze ( theme = * raisin ) DONE

### B.2.2 EXEMPLAR MATCHING

For matching, we use essentially the same heuristics as we used for CFQ with slightly adjusted parameters. In particular, we only look at the inner-most subclause when selecting exemplars. This is sufficient because the handling of the nested subclauses and nested prepositional clauses is demonstrated with a static prompt that we provide in addition to the dynamically selected exemplars (see Section B.3 below).

If the inner-most subclause has a subject but no object (e.g., "a donut snapped"), we select the exemplar for the verb of this subclause, which is either unaccusative or unergartive. This tells the model whether the subject should be annotated as the agent or the theme (e.g., since "snap" is unaccusative "a donate" is annotated as the theme). If the inner-most subclause has both a subject and an object, we select 3 exemplars corresponding to the subclause structure.

**Example 1.** In our first example "James said that a manager liked that Aiden appreciated that Emily believed that the girl was posted a cake beside a table by Olivia .", the inner-most subclause is "the girl was posted a cake by Olivia", which means that we select the following three exemplars:

Q: (The * frog) (was mailed [mail]) (the * ball) by (the * child)
A: PARSE: mail ( recipient = * frog , theme = * ball , agent = * child ) DONE

Q: (The * dog) (was given [give]) (the * cookie) by (the * duck)
A: PARSE: give ( recipient = * dog , theme = * cookie , agent = * duck ) DONE

Q: (The * prince) (was passed [pass]) (the * box) by (the * president)
A: PARSE: pass ( recipient = * prince , theme = * box , agent = * president ) DONE

**Example 2.** In our second example "The boy shortened the donut beside the bed in the car in the garden in the can on the tree .", the inner-most subclause is "the boy shortened the donut", which means that we select the following three exemplars:

Q: (The * pig) (liked [like]) (the * zebra)
A: PARSE: like ( agent = * pig , theme = * zebra ) DONE

Q: (The * baby) (missed [miss]) (the * cake)
A: PARSE: miss ( agent = * baby , theme = * cake ) DONE

Q: (The * creature) (appreciated [appreciate]) (the * present)
A: PARSE: appreciate ( agent = * creature , theme = * present ) DONE

### B.3 COGS SOLUTION: DETAILS AND PROMPTS

As we discussed in Section 3.3, we linearize the composition tree into a sequence of sentences that correspond to increasingly complex subproblems. Like with CFQ (see Section A.3, this linearization is performed by a walk over the parse tree to obtain a linear sequence of well-formed sentences. For each sentence, we then query the language model using a prompt that consists of three parts.

1. Static prompt context illustrating the composition of subclauses and prepositional phrases

2. Dynamically selected exemplars as additional context

3. Sequential subproblems.

These parts are detailed throughout the rest of this section.

### B.3.1 Part 1: Static prompt context

The compositions of subclauses and prepositional phrases are translated very systematically. Furthermore, because of the sequential prompting, the model only needs to perform one step of composition at a time. As a consequence, it is sufficient to demonstrate this behavior with the following static prompt context that is used for all inputs. Noe that this prompt does not contain any nesting beyond level 2. Note that we again use a parsed version of the natural language sentences, which contain additional parentheses and other annotations that make it easier for the model to perform the translation.

Q: (the * song) (needed [need]) (to help)
A: PARSE: need ( agent = * song , xcomp = help ( agent = * song ) ) DONE

Q: (The * camera) (hoped [hope]) (to run)
A: PARSE: hope ( agent = * camera , xcomp = run ( agent = * camera ) ) DONE

Q: (the * atom) (wanted [want]) (to hunt)
A: PARSE: wanted ( agent = * atom , xcomp = hunt ( agent = * atom ) ) DONE

Q: (a helmet) (preferred [prefer]) (to cook)
A: PARSE: prefer ( agent = helmet , xcomp = cook ( agent = helmet ) ) DONE

Q: (Fred) (expected [expect]) (to paint)
A: PARSE: expect ( agent = Fred , xcomp = paint ( agent = Fred ) ) DONE

Q: (A camera) (intended [intend]) (to clean)
A: PARSE: intend ( agent = camera , xcomp = clean ( agent = camera ) ) DONE

Q: ((A camera) (beside) (the * song)) (intended [intend]) (to clean)
A: PARSE: intend ( agent = camera ( nmod . beside = * song ) , xcomp = clean ( agent = camera ( nmod . beside = * song ) ) ) DONE

Q: (Lucas) (liked [like]) (to eat)
A: PARSE: like ( agent = Lucas , xcomp = eat ( agent = Lucas ) ) DONE

Q: ((Lucas) (in) (the * helmet)) (liked [like]) (to eat)
A: PARSE: like ( agent = Lucas ( nmod . in = * helmet ) , xcomp = eat ( agent = Lucas ( nmod . in = * helmet ) ) ) DONE

Q: (the * atom) (returned [return]) (a camera) (Louise)
A: PARSE: return ( agent = * atom , recipient = camera , theme = Louise ) DONE

Q: ((the * atom) (beside) (the * rino)) (returned [return]) (a camera) (Louise)
A: PARSE: return ( agent = * atom ( nmod . beside = * rino ) , recipient = camera , theme = Louise ) DONE

Q: (Liam) (returned [return]) (Noah) to (the * helmet)
A: PARSE: return ( agent = Liam , theme = Noah , recipient = * helmet ) DONE

Q: (Liam) (returned [return]) (Noah) to ((the * helmet) (on) (the * atom))
A: PARSE: return ( agent = Liam , theme = Noah , recipient = * helmet ( nmod . on = * atom ) ) DONE

Q: (the * rino) (was returned [return]) (a helmet)
A: PARSE: return ( recipient = * rino , theme = helmet ) DONE

Q: (the * rino) (was returned [return]) ((a helmet) (on) (the * camera))
A: PARSE: return ( recipient = * rino , theme = helmet ( nmod . on = * camera ) ) DONE

Q: (Zoey) (was returned [return]) (the * song) by (the * helmet)

A: PARSE: return ( recipient = Zoey , theme = * song , agent = * helmet ) DONE

Q: (Zoey) (was returned [return]) ((the * song) (in) (the * camera)) by (the * helmet)
A: PARSE: return ( recipient = Zoey , theme = * song ( nmod . in = * camera ) , agent = * helmet ) DONE

Q: (Anna) (was returned [return]) to (the * TV)
A: PARSE: return ( theme = Anna , recipient = * tv ) DONE

Q: (Anna) (was returned [return]) to ((the * TV) (beside) (the * rino))
A: PARSE: return ( theme = Anna , recipient = * tv ( nmod . beside = * rino ) ) DONE

Q: (A atom) (was passed [pass]) to (Nicole) by (a camera)
A: PARSE: pass ( theme = atom , recipient = Nicole , agent = camera ) DONE

Q: (A atom) (was passed [pass]) to (Nicole) by ((a camera) (in) (the * song))
A: PARSE: pass ( theme = atom , recipient = Nicole , agent = camera ( nmod . in = * song ) ) DONE

Q: (A atom) (was wired [wire]) (a TV) by (the * song)
A: PARSE: wire ( recipient = atom , theme = tv , agent = * song) DONE

Q: (A atom) (was wired [wire]) ((a TV) (on) (the * camera)) by (the * song)
A: PARSE: wire ( recipient = atom , theme = tv ( nmod . on = * camera ) , agent = * song ) DONE

Q: (Noah) (hoped [hope]) that ((a atom) (was wired [wire]) ((a TV) (on) (the * camera)) by (the * song))
A: PARSE: hope ( agent = Noah , ccomp = wire ( recipient = atom , theme = tv ( nmod . on = * camera ) , agent = * song ) ) DONE

Q: (a song) (ate [eat]) (the * atom)
A: PARSE: eat ( agent = song , theme = * atom ) DONE

Q: ((a song) (in) (the * camera)) (ate [eat]) (the * atom)
A: PARSE: eat ( agent = song ( nmod . in = * camera ) , theme = * atom ) DONE

Q: (The * rino) (valued [value]) that (((a song) (in) (the * camera)) (ate [eat]) (the * atom))
A: PARSE: value ( agent = * rino , ccomp = eat ( agent = song ( nmod . in = * camera ) , theme = * atom ) ) DONE

Q: ((The * rino (beside) (the * helmet)) (valued [value]) that (((a song) (in) (the * camera)) (ate [eat]) (the * atom))
A: PARSE: value ( agent = * rino ( nmod . beside = * helmet ) , ccomp = eat ( agent = song ( nmod . in = * camera ) , theme = * atom ) ) DONE

### B.3.2   PART 2: DYNAMICALLY SELECTED EXEMPLARS

After the constant prompt context, we add as additional context the exemplars that we dynamically selected for the given input using the process described in Section B.2. We provide these exemplars for both of our runnning examples in Section B.2.2 and do not repeat them here.

### B.3.3   PART 3: SEQUENTIAL SUBPROBLEMS

After providing the static and dynamic prompt context, we perform the sequential prompting. We start by appending to the prompt the simplest subproblem and send it as a request to the language model. We then append the model's reply before we append the next subproblem and make another request to the language model. This is done until we obtain the result for the final subproblem, which corresponds to the solution of the original problem.

**Example 1.** For the first example ("James said that a manager liked that Aiden appreciated that Emily believed that the girl was posted a cake beside a table by Olivia ."), this part of the prompt looks as follows. Note that the not bolded text corresponds to answers provided by the model.

**Q: (the * girl) (was posted [post]) (a cake) by (Olivia)**
A: PARSE: post ( recipient = * girl , theme = cake , agent = Olivia ) DONE

**Q: (the * girl) (was posted [post]) ((a cake) (beside) (a table)) by (Olivia)**
A: PARSE: post ( recipient = * girl , theme = cake ( nmod . beside = table ) , agent = Olivia ) DONE

**Q: (Emily) (believed [believe]) that ((the * girl) (was posted [post]) ((a cake) (beside) (a table)) by (Olivia))**
A: PARSE: believe ( agent = Emily , ccomp = post ( recipient = * girl , theme = cake ( nmod . beside = table ) , agent = Olivia ) ) DONE

**Q: (Aiden) (appreciated [appreciate]) that ((Emily) (believed [believe]) that ((the * girl) (was posted [post]) ((a cake) (beside) (a table)) by (Olivia)))**
A: PARSE: appreciate ( agent = Aiden , ccomp = believe ( agent = Emily , ccomp = post ( recipient = * girl , theme = cake ( nmod . beside = table ) , agent = Olivia ) ) ) DONE

**Q: (a manager) (liked [like]) that ((Aiden) (appreciated [appreciate]) that ((Emily) (believed [believe]) that ((the * girl) (was posted [post]) ((a cake) (beside) (a table)) by (Olivia))))**
A: PARSE: like ( agent = manager , ccomp = appreciate ( agent = Aiden , ccomp = believe ( agent = Emily , ccomp = post ( recipient = * girl , theme = cake ( nmod . beside = table ) , agent = Olivia ) ) ) ) DONE

**Q: (James) (said [say]) that ((a manager) (liked [like]) that ((Aiden) (appreciated [appreciate]) that ((Emily) (believed [believe]) that ((the * girl) (was posted [post]) ((a cake) (beside) (a table)) by (Olivia)))))**
A: PARSE: say ( agent = James , ccomp = like ( agent = manager , ccomp = appreciate ( agent = Aiden , ccomp = believe ( agent = Emily , ccomp = post ( recipient = * girl , theme = cake ( nmod . beside = table ) , agent = Olivia ) ) ) ) ) DONE

**Example 2.** For the second example ("The boy shortened the donut beside the bed in the car in the garden in the can on the tree ."), the sequential part of the prompt looks as follows.

**Q: (The * boy) (shortened [shorten]) (the * donut)**
A: PARSE: shorten ( agent = * boy , theme = * donut ) DONE

**Q: (The * boy) (shortened [shorten]) ((the * donut) (beside) (the * bed))**
A: PARSE: shorten ( agent = * boy , theme = * donut ( nmod . beside = * bed ) ) DONE

**Q: (The * boy) (shortened [shorten]) ((the * donut) (beside) ((the * bed) (in) (the * car)))**
A: PARSE: shorten ( agent = * boy , theme = * donut ( nmod . beside = * bed ( nmod . in = * car ) ) ) DONE

**Q: (The * boy) (shortened [shorten]) ((the * donut) (beside) ((the * bed) (in) ((the * car) (in) (the * garden))))**
A: PARSE: shorten ( agent = * boy , theme = * donut ( nmod . beside = * bed ( nmod . in = * car ( nmod . in = * garden ) ) ) ) DONE

**Q: (The * boy) (shortened [shorten]) ((the * donut) (beside) ((the * bed) (in) ((the * car) (in) ((the * garden) (in) (the * can)))))**
A: PARSE: shorten ( agent = * boy , theme = * donut ( nmod . beside = * bed ( nmod . in = * car ( nmod . in = * garden ( nmod . in = * can ) ) ) ) ) DONE

**Q: (The * boy) (shortened [shorten]) ((the * donut) (beside) ((the * bed) (in) ((the * car) (in) ((the * garden) (in) ((the * can) (on) (the * tree))))))**
A: PARSE: shorten ( agent = * boy , theme = * donut ( nmod . beside = * bed ( nmod . in = * car ( nmod . in = * garden ( nmod . in = * can ( nmod . on = * tree ) ) ) ) ) ) DONE

# C   CHAIN-OF-THOUGHT PROMPTING FOR CFQ

This section includes the chain-of-thought prompts introduced in Section 4. A prompt in chain-of-thought format includes intermediate steps that are processed before predicting the final answer. Although the exemplars include these intermediate steps already, the model must predict the steps for new input. Also, it is assumed that the exemplar pool has not previously been annotated with chain-of-thought, so a procedure to bootstrap these chain-of-thought (also called rationale) is required. In practice, we can do this by annotating a small set of exemplars (in our case 5), then using these to teach the model to predict new chain-of-thought through prompting. The prompt we use for bootstrapping is in Appendix C.1.

For evaluation, we use prompts including a mix of exemplars, some in chain-of-thought format, and the rest in vanilla format, this allows us to keep the prompt relatively short, since chain-of-thought can be verbose, often the same length or longer than the original exemplar. An example chain-of-thought prompt uused in evaluation is shown in Appendix C.2. Note, the vanilla few-shot prompts we use are similar but only includes exemplars in the vanilla format.

We only evaluate chain-of-thought and vanilla few-shot prompting against CFQ, not COGS.

## C.1   BOOTSTRAP PROMPT

This is a hybrid prompt containing 15 vanilla input/output exemplars selected by bag-of-words similarity with the input, and a static set of 5 exemplars manually annotated in the chain-of-thought format. This prompt is used to generate new chain-of-thought for the rest of the exemplar pool. An example is shown below in reduced font size, since the chain-of-thought can be verbose.

```
# SQL Dataset:

input: Who was a actor that M2 starred and M3 was directed by

output: SELECT DISTINCT ?x0 WHERE { ?x0 a actor . ?x0 a person . ?x0 starred_in M2 . ?x0 directed M3 }

input: Were M2 and M4 edited by a male sibling of M0 and directed by M3

output: SELECT count(*) WHERE { ?x0 has_gender male . ?x0 sibling_of M0 . M2 directed_by M3 . M2 edited_by ?x0
    . M4 directed_by M3 . M4 edited_by ?x0 }

input: Who was a Swedish actor that M3 married and M4 's producer influenced

output: SELECT DISTINCT ?x0 WHERE { ?x0 a actor . ?x0 a person . ?x0 influenced_by ?x1 . ?x0 has_nationality
    Swedish . ?x0 married_to M3 . ?x1 produced M4 }

input: What was produced and directed by a cinematographer 's Chinese sibling

output: SELECT DISTINCT ?x0 WHERE { ?x0 directed_by ?x1 . ?x0 produced_by ?x1 . ?x1 has_nationality Chinese .
    ?x1 sibling_of ?x2 . ?x2 a cinematographer }

input: Who was a film director that M2 was influenced by and a founder of M3 and M4 was influenced by

output: SELECT DISTINCT ?x0 WHERE { ?x0 a film_director . ?x0 a person . ?x0 influenced ?x1 . ?x0 influenced
    M2 . ?x1 founded M3 . ?x1 founded M4 }

input: Did M2 and M4 influence M3 , influence M0 's child , and influence a cinematographer 's sibling 's
    spouse

output: SELECT count(*) WHERE { ?x0 child_of M0 . ?x1 married_to ?x2 . ?x2 sibling_of ?x3 . ?x3 a
    cinematographer . M2 influenced ?x0 . M2 influenced ?x1 . M2 influenced M3 . M4 influenced ?x0 . M4
    influenced ?x1 . M4 influenced M3 }

input: Who married , influenced , and was influenced by a cinematographer that M2 was directed by and starred

output: SELECT DISTINCT ?x0 WHERE { ?x0 a person . ?x0 influenced ?x1 . ?x0 influenced_by ?x1 . ?x0 married_to
    ?x1 . ?x1 a cinematographer . ?x1 starred_in M2 . ?x1 directed M2 }

input: What was written by and directed by a female sibling of a cinematographer of M1

output: SELECT DISTINCT ?x0 WHERE { ?x0 directed_by ?x1 . ?x0 written_by ?x1 . ?x1 has_gender female . ?x1
    sibling_of ?x2 . ?x2 cinematographer_of M1 }

input: Who influenced , married , and was influenced by a actor that edited M2 and directed M3

output: SELECT DISTINCT ?x0 WHERE { ?x0 a person . ?x0 influenced ?x1 . ?x0 influenced_by ?x1 . ?x0 married_to
    ?x1 . ?x1 a actor . ?x1 directed M3 . ?x1 edited M2 }

input: Was M2 directed by M3 and M4 and written by a male sibling of M0

output: SELECT count(*) WHERE { ?x0 has_gender male . ?x0 sibling_of M0 . M2 directed_by M3 . M2 directed_by
    M4 . M2 written_by ?x0 }
```

```
input: Who was influenced by M1 , influenced by M4 's producer , cinematographer , and director , and
    influenced by M2 and M3

output: SELECT DISTINCT ?x0 WHERE { ?x0 a person . ?x0 influenced_by ?x1 . ?x0 influenced_by M1 . ?x0
    influenced_by M2 . ?x0 influenced_by M3 . ?x1 cinematographer_of M4 . ?x1 directed M4 . ?x1 produced M4
    }

input: Were M2 , M3 , M4 , M5 , and M6 influenced by a Spanish spouse of M1

output: SELECT count(*) WHERE { ?x0 has_nationality Spanish . ?x0 married_to M1 . M2 influenced_by ?x0 . M3
    influenced_by ?x0 . M4 influenced_by ?x0 . M5 influenced_by ?x0 . M6 influenced_by ?x0 }

input: Were M2 and M3 directed by and edited by a cinematographer 's French parent

output: SELECT count(*) WHERE { ?x0 parent_of ?x1 . ?x0 has_nationality French . ?x1 a cinematographer . M2
    directed_by ?x0 . M2 edited_by ?x0 . M3 directed_by ?x0 . M3 edited_by ?x0 }

input: Who was a film producer whose sibling directed M3 and edited M2

output: SELECT DISTINCT ?x0 WHERE { ?x0 a film_producer . ?x0 a person . ?x0 sibling_of ?x1 . ?x1 directed M3
    . ?x1 edited M2 }

## Example Parsings:

Query: What was executive produced by , directed by , and edited by M1 's male spouse
Query Type: What => DISTINCT
There is an entity (?x0) => ?x0 a entity
?x0 is executive produced by M1's male spouse => ?x0 executive_produced_by ?x1, ?x1 has_gender male, ?x1
    married_to M1
?x0 is directed by M1's male spouse => ?x0 directed_by ?x1
?x0 is edited by M1's male spouse => ?x0 edited_by ?x1

So the parse of this query is:
Parse: SELECT DISTINCT ?x0 WHERE { ?x0 directed_by ?x1 . ?x0 edited_by ?x1 . ?x0 executive_produced_by ?x1 . ?
    x1 has_gender male . ?x1 married_to M1 }

Query: Were M0 , M4 , M5 , M6 , and M7 directed by M3 , executive produced by M1 , and written by M2
Query Type: were/was => count(*)
M0 is directed by M3 => M0 directed_by M3
M0, M4, M5, M6 is executive produced by M1 => M0 executive_produced_by M1, M4 executive_produced_by M1, M5
    executive_produced_by M1, M6 executive_produced_by M1, M7 executive_produced_by M1
M0, M4, M5, M6 is written by M2 => M0 written_by M2, M4 written_by M2, M5 written_by M2, M6 written_by M2
M0, M4, M5, M6 is directed by M3 => M0 directed_by M3, M4 directed_by M3, M5 directed_by M3, M6 directed_by M3

So the parse of this query is:
Parse: SELECT count(*) WHERE { M0 directed_by M3 . M0 executive_produced_by M1 . M0 written_by M2 . M4
    directed_by M3 . M4 executive_produced_by M1 . M4 written_by M2 . M5 directed_by M3 . M5
    executive_produced_by M1 . M5 written_by M2 . M6 directed_by M3 . M6 executive_produced_by M1 . M6
    written_by M2 . M7 directed_by M3 . M7 executive_produced_by M1 . M7 written_by M2 }

Query: Was a Japanese screenwriter whose parent played M0 and M1 M2
Query Type: was/were => count(*)
There is a Japanese screenwriter (?x0) => ?x0 a writer, ?x0 has_nationality Japanese
?x0's parent is ?x1 => ?x0 child_of ?x1
?x1 played M0 and M1 => ?x1 portrayed M0, ?x1 portrayed M1

So the parse of this query is:
Parse: SELECT count(*) WHERE { ?x0 portrayed M0 . ?x0 portrayed M1 . M2 a writer . M2 has_nationality Japanese
    . M2 child_of ?x0 }

Query: Was M1 produced by M0 's editor and art director , produced by M3 and M4 , and distributed by M2
Query Type: was/were => count(*)
There is an editor (?x0) of M0 => ?x0 edited M0
?x0 is art director of M0 => ?x0 art_directed M0
M1 is distributed by M2 => M1 distributed_by M2
M1 is produced by ?x0 => M1 produced_by ?x0
M1 is produced by M3 => M1 produced_by M3
M1 is produced by M4 => M1 produced_by M4

So the parse of this query is:
Parse: SELECT count(*) WHERE { ?x0 edited M0 . ?x0 art_directed M0 . M1 distributed_by M2 . M1 produced_by ?x0
    . M1 produced_by M3 . M1 produced_by M4 }

Query: Was a film producer 's child founded by M0 and M1
Query Type: was/were => count(*)
There is a child (?x0) of film producer (?x1) => ?x0 child_of ?x1, ?x1 a film_producer
?x0 is founded by M0 and M1 => ?x0 founded_by M0, ?x0 founded_by M0

So the parse of this query is:
Parse: SELECT count(*) WHERE { ?x0 founded_by M0 . ?x0 founded_by M1 . ?x0 child_of ?x1 . ?x1 a film_producer
    }

Query: Who was a French cinematographer whose sibling directed M3 and M4
Query Type:
```

## C.2 Prediction Prompt

This is an example of the chain-of-thought prompt used to predict semantic parse output on evaluation data. It includes 10 exemplars in the vanilla input/output format, then 5 exemplars in the chain-of-thought format. The example is shown below in reduced font size, since the chain-of-thought can be verbose.

```
input: Were M2 and M3 written by M0 's producer 's employer 's founder and edited by a screenwriter
output: SELECT count(*) WHERE { ?x0 founded ?x1 . ?x1 employed ?x2 . ?x2 produced M0 . ?x3 a writer . M2
    edited_by ?x3 . M2 written_by ?x0 . M3 edited_by ?x3 . M3 written_by ?x0 }

input: Was M2 executive produced by M3 , edited by a film editor , and edited by M0 's producer , executive
    producer , and writer
output: SELECT count(*) WHERE { ?x0 executive_produced M0 . ?x0 produced M0 . ?x0 wrote M0 . ?x1 a film_editor
     . M2 edited_by ?x0 . M2 edited_by ?x1 . M2 executive_produced_by M3 }

input: Was M2 founded by a screenwriter 's French parent and founded by M3
output: SELECT count(*) WHERE { ?x0 parent_of ?x1 . ?x0 has_nationality French . ?x1 a writer . M2 founded_by
    ?x0 . M2 founded_by M3 }

input: Was a French film producer that was employed by M2 M0
output: SELECT count(*) WHERE { M0 a film_producer . M0 employed_by M2 . M0 has_nationality French }

input: Was M2 executive produced by and written by a screenwriter 's Italian sibling
output: SELECT count(*) WHERE { ?x0 has_nationality Italian . ?x0 sibling_of ?x1 . ?x1 a writer . M2
    executive_produced_by ?x0 . M2 written_by ?x0 }

input: Were M2 and M5 founded by M3 and M4 , founded by a screenwriter , and founded by M1 's executive
    producer
output: SELECT count(*) WHERE { ?x0 a writer . ?x1 executive_produced M1 . M2 founded_by ?x0 . M2 founded_by ?
    x1 . M2 founded_by M3 . M2 founded_by M4 . M5 founded_by ?x0 . M5 founded_by ?x1 . M5 founded_by M3 . M5
     founded_by M4 }

input: Did M2 marry a screenwriter , marry M3 , and influence M0 's producer
output: SELECT count(*) WHERE { ?x0 produced M0 . ?x1 a writer . M2 influenced ?x0 . M2 married_to ?x1 . M2
    married_to M3 }

input: Was M1 produced by M2 , edited by a film producer 's employee and founder , and directed by M3
output: SELECT count(*) WHERE { ?x0 founded ?x1 . ?x0 employed_by ?x1 . ?x1 a film_producer . M1 directed_by
    M3 . M1 edited_by ?x0 . M1 produced_by M2 }

input: Was M1 executive produced by a producer of M0 's prequel and written by M2
output: SELECT count(*) WHERE { ?x0 produced ?x1 . ?x1 has_sequel M0 . M1 executive_produced_by ?x0 . M1
    written_by M2 }

input: Was M1 employed by M2 and employed by a film 's producer and distributor
output: SELECT count(*) WHERE { ?x0 distributed ?x1 . ?x0 produced ?x1 . ?x1 a film . M1 employed_by ?x0 . M1
    employed_by M2 }

Query: Was a screenwriter 's British parent 's parent M2
Query Type: was/were => count(*)
There is a screenwriter (?x0) => ?x0 a writer
?x0's parent is ?x1 => ?x0 parent_of ?x1
?x1 is British => ?x1 has_nationality British
?x1's parent is M2 => M2 parent_of ?x1

So the parse of this query is:
Parse: SELECT count(*) WHERE { ?x0 parent_of ?x1 . ?x0 has_nationality British . ?x1 a writer . M2 parent_of ?
    x0 }

Query: Was M2 edited by a Swedish film producer 's spouse and written by M3
Query Type: was/were => count(*)
There is a Swedish film producer (?x0) => ?x0 a film_producer, ?x0 has_nationality Swedish
?x0's spouse is ?x1 => ?x0 married_to ?x1
M2 is edited by ?x1 => M2 edited_by ?x1
M2 is written by M3 => M2 written_by M3

So the parse of this query is:
Parse: SELECT count(*) WHERE { ?x0 married_to ?x1 . ?x1 a film_producer . ?x1 has_nationality Swedish . M2
    edited_by ?x0 . M2 written_by M3 }

Query: Was M2 a film written by M4 and directed by M0 's male executive producer
Query Type: was/were => count(*)
There is a male executive producer (?x0) of M0 => ?x0 executive_produced M0, ?x0 has_gender male
M2 is directed by ?x0 => M2 directed_by ?x0
M2 is written by M4 => M2 written_by M4
M2 is a film => M2 a film
```

```
So the parse of this query is:
Parse: SELECT count(*) WHERE { ?x0 executive_produced M0 . ?x0 has_gender male . M2 a film . M2 directed_by ?
     x0 . M2 written_by M4 }

Query: Was a film producer influenced by a costume designer 's sibling and influenced by M1 and M2
Query Type: was/were => count(*)
There is a film producer (?x0) => ?x0 a film_producer
?x0 is influenced by a costume designer's sibling (?x1) => ?x0 influenced_by ?x1, ?x1 sibling_of ?x2, ?x2 a
     costume_designer
?x0 is influenced by M1 => ?x0 influenced_by M1
?x0 is influenced by M2 => ?x0 influenced_by M2

So the parse of this query is:
Parse: SELECT count(*) WHERE { ?x0 a film_producer . ?x0 influenced_by ?x1 . ?x0 influenced_by M1 . ?x0
     influenced_by M2 . ?x1 sibling_of ?x2 . ?x2 a costume_designer }

Query: Was M3 produced by a Mexican film producer and produced by M2 's star 's spouse 's sibling
Query Type: was/were => count(*)
There is a Mexican film producer (?x0) => ?x0 a film_producer, ?x0 has_nationality Mexican
?x0 is produced by M2's star's spouse's sibling (?x1) => ?x1 sibling_of ?x2, ?x2 married_to ?x3, ?x3
     starred_in M2, M3 produced_by ?x0, M3 produced_by ?x1

So the parse of this query is:
Parse: SELECT count(*) WHERE { ?x0 a film_producer . ?x0 has_nationality Mexican . ?x1 sibling_of ?x2 . ?x2
     married_to ?x3 . ?x3 starred_in M2 . M3 produced_by ?x0 . M3 produced_by ?x1 }

Query: Was a screenwriter M2 's French producer
Query Type:
```

| | |
|---|---|
| ns:organization.organization.companies_acquired/ns:business.acquisition.company_acquired | acquired |
| ns:organization.organization.acquired_by/ns:business.acquisition.acquiring_company | acquired_by |
| ns:film.actor.film/ns:film.performance.film | starred_in |
| ns:film.film_art_director.films_art_directed | art_directed |
| ns:film.film.film_art_direction_by | art_direction_by |
| ns:people.person.parents\|ns:fictional_universe.fictional_character.parents—ns:organization.organization.parent/ns:orga | child_of |
| ns:film.cinematographer.film | cinematographer_of |
| ns:film.film.cinematography | cinematography_by |
| ns:film.film.costumer_designer.costume_design_for_film | costume_designed |
| ns:film.film.costume_design_by | costume_designed_by |
| ns:film.director.film | directed |
| ns:film.film.directed_by | directed_by |
| ns:film.film_distributor.films_distributed/ns:film.film_film_distributor_relationship.film | distributed |
| ns:film.film.distributors/ns:film.film_film_distributor_relationship.distributor | distributed_by |
| ns:film.editor.film | edited |
| ns:film.film.edited_by | edited_by |
| ns:business.employer.employees/ns:business.employment_tenure.person | employed |
| ns:people.person.employment_history/ns:business.employment_tenure.company | employed_by |
| ns:film.producer.films_executive_produced | executive_produced |
| ns:film.film.executive_produced_by | executive_produced_by |
| ns:organization.organization_founder.organizations_founded | founded |
| ns:organization.organization.founders | founded_by |
| fis:people.person.gender | gender_of |
| ns:film.actor.film/ns:film.performance.character | portrayed |
| ns:people.person.gender | has_gender |
| ns:people.person.nationality | has_nationality |
| ns:film.film.prequel | has_prequel |
| ns:film.film.sequel | has_sequel |
| ns:influence.influence_node.influenced | influenced |
| ns:influence.influence_node.influenced_by | influenced_by |
| ns:people.person.spouse_s/ns:people.marriage.spouse\|ns:fictional_universe.fictional_character.married_to/ns:fictiona | married_to |
| fis:people.person.nationality | nationality_of |
| ns:people.person.children\|ns:fictional_universe.fictional_character.children\|ns:organization.organization.child/ns:org | parent_of |
| ns:film.producer.film\|ns:film.production_company.films | produced |
| ns:film.film.produced_by\|ns:film.film.production_companies | produced_by |
| ns:people.person.sibling_s/ns:people.sibling_relationship.sibling\|ns:fictional_universe.fictional_character.siblings | sibling_of |
| ns:film.film.starring/ns:film.performance.actor | starred |
| ns:film.film.written_by | written_by |
| ns:film.writer.film | wrote |
| ns:film.actor | actor |
| ns:film.film_art_director | art_director |
| ns:film.cinematographer | cinematographer |
| ns:film.film_costumer_designer | costume_designer |
| ns:film.director | film_director |
| ns:film.editor | film_editor |
| ns:business.employer | employer |
| ns:fictional_universe.fictional_character | fictional_character |
| ns:film.film | film |
| ns:film.film_distributor | film_distributor |
| ns:people.person | person |
| ns:film.producer | film_producer |
| ns:film.production_company | production_company |
| ns:film.writer | writer |

Table 4: Mapping of freebase IDs (truncated to 120 characters) to human-readable strings. We apply this to CFQ to make the task more feasible for prompting.

# D    DATA PREPARATION, EVALUATION, AND HYPERPARAMETERS

## D.1    CFQ PROCESSING AND EVALUATION

To make the CFQ benchmark more appropriate for processing with large language models, we use the processing steps and evaluation method detailed below. We verified that these steps do not alter the performance for fully supervised methods by reproducing experiments with T5-base. The results match previously published results within 1-2 points on MCD1, MCD2, and MCD3.

### D.1.1    CFQ PREPROCESSING

To prepare the CFQ data we apply two preprocessing steps:

1. We replace excessively long freebase identifiers with human-readable strings (see Table 4).

2. We strip FILTER statements because they always appear with the "sibling_of" and "married_to" properties, and would be trivial to add to the output after prediction (i.e., they can be considered to be part of the abbreviated property). For example, if "?x0 married_to M0" appeared, then so would "FILTER(?x0 != M0)". Essentially, one cannot be married to themselves, nor a sibling of themselves.

### D.1.2    CFQ POSTPROCESSING AND EVALUATION

For CFQ, we apply the following post-processing to both gold and predicted semantic parses.

1. *Clause sanitization*: We discard any malformed clause. To measure well-formedness we check all of the following. (a) A clause should have 3 white-space separated tokens; (b) A clause should either have "a" as its second token or a string, so symbols such as "=" can be safely discarded.

2. *Inverse properties*: For properties that have an official inverse property in Freebase (e.g. "directed" vs. "directed_by"), we deterministically pick one of its variants and flip the arguments if needed. However, note that in some error cases the model will predict "sequel_of", which does not get corrected, since "sequel_of" is not in the original property vocabulary, only "has_sequel" and "has_prequel".

3. *Argument ordering*: For clauses that are bidirectional ("sibling_of", "married_to"), we sort the arguments alphabetically.

4. *Statement ordering*: Since order does not matter in SPARQL statements, we sort statements alphabetically.

5. *Variable normalization*: Since variable labels are arbitrary (?x0, ?x1, etc.) in SPARQL, we re-label the variables so that they appear in increasing order. E.g., "SELECT ?x1 { ?x2 directed ?x1 . ?x1 influenced ?x0 }" gets converted to "SELECT ?x0 { ?x1 directed ?x0 . ?x0 influenced ?x2 }". We alternate running *variable normalization* and *statement ordering* until no change is detected, since re-labeling variables might impact the sort order of clauses.

6. *Stripping of implied types*: CFQ is constructed such that types that are directly implied by a accompanied relation are dropped from the SPARQL even if these types are explicitly mentioned in the natural language question. For example, the clause "?x0 a actor" is dropped from the translation of the question "Did a male actor play M0 and play M1" because the relation "?x0 portrayed M0" implies that ?x0 has type actor. For a pretrained language model, this is quite unnatural and hurts accuracy, even though keeping the type leads to a SPARQL query that is semantically equivalent. We therefore strip implied types when they are predicted by the language model.

After completing these steps, we do the standard approach of measuring accuracy using exact string match.

## D.2 COGS POSTPROCESSING AND EVALUATION

For longer outputs, the language model sometimes fails to match closing parentheses. When this happens, we add closing parentheses at the end of output until all opening parenthesis are matched. This trivial fix improved exact match accuracy on COGS generalization test set from 97.8% to 99.2%.

It is worth noting that 50 examples in the original COGS generalization test set are mislabeled (the authors have since released a new version). We evaluate using the original data in order to compare with previous work. As one would not expect a model to accurately predict the idiosyncrasies associated with the mislabeled data, an upper bound on performance for the generalization test should be about 99.7% accuracy.

## D.3 HYPERPARAMETERS

There are two sets of hyperparameters: those for prompts and those for generation. We performed initial minimal hyperparameter tuning using a 100-sentence subset of the validation data.

### D.3.1 PROMPT HYPERPARAMETERS

**Dynamic Least-to-Most**  Described in Section 3.

- Number of Static Exemplars = 12 for CFQ, 28 for COGS
- Number of Dynamic Exemplars = 4-35 for CFQ, between 1-3 for COGS (these are determined automatically based on the decomposition tree)
- Number of Exemplar Lists = 1
- Number of Generations per List = 1
- Generation Mode = Greedy

**Chain-of-Thought**  Selects exemplars according to bag-of-words similarity (Section 4.2), but to be effective, at least some of the exemplars must use rationale. To keep the prompt compact, we use a hybrid approach where 5 exemplars are in chain-of-thought format and the rest are vanilla.

As the original data does not have alignments, we need to manually create some chain-of-thought exemplars, which we can then use to generate chain-of-thought for the full exemplar pool (Zelikman et al., 2022). In our case, we manually labeled 5 sentences with chain-of-thought, then appended these to the 15 exemplars we already retrieve for each sentence. We did this to predict chain-of-thought for the exemplar pool, and discarded any chain-of-thought that did not produce the correct semantic parse. 884 out of 1000 successfully predicted the correct semantic parse. When constructing chain-of-thought prompts, we only represent an exemplar with chain-of-thought if it succeeded in this previous step, otherwise we use its vanilla format.

- Bootstrap Pool Size = 5
- Number of Exemplars per List ($k$) = 15
- Number of Exemplar Lists ($n$) = 4
- Number of Generations per List ($s$) = 4
- Generation Mode = Sample
- Number of Chain-of-Thought Exemplars = 5 (the other 10 are in vanilla format)

**Vanilla Few-Shot**  Vanilla few-shot is a simple input/output exemplar-based prompt. The exemplars are chosen identically as chain-of-thought.

- Number of Exemplars per List ($k$) = 15
- Number of Exemplar Lists ($n$) = 4
- Number of Generations per List ($s$) = 4
- Generation Mode = Sample

### D.3.2 GENERATION HYPERPARAMETERS

We did not require any extensive search over these hyperparamters. We tried reasonable settings based on previously published works, and the only real toggle here is temperature, which wasn't used at all by dynamic least-to-most.

- Model = `code-davinci-002`
- Temperature = 0.7 when sampling, or 0.0 when greedy.
- Top-P = 1.0, Presence Penalty = 0, Frequency Penalty = 0

## E ADDITIONAL ANALYSIS

### E.1 INITIAL PROMPTING ATTEMPTS FOR CFQ

We initially tried various prompting attempts for CFQ where we did not provide any examples as part of the context. Instead, we provided an instruction along the lines of, "Translate English to SPARQL". While the language model was still able to produce SPARQL-like output, it scored 0% accuracy on CFQ. This indicates that the language model was exposed to SPARQL in pretraining but has not seen or memorized specific CFQ examples.

```
# Translate English to SPARQL.

English: What did a production company 's Japanese founder direct , edit ,
    and executive produce
SPARQL:
```
SELECT ?director ?editor ?executive_producer
WHERE {
  ?director a dbo:FilmDirector .
  ?director dbo:nationality dbr:Japan .
  ?director dbo:birthPlace dbr:Japan .
  ?director dbo:birthDate ?birthDate .
  FILTER (?birthDate < "1940-01-01T00:00:00Z"^^xsd:dateTime)
  ?director dbo:film ?film .
  ?film dbo:editor ?editor .
  ?film dbo:executiveProducer ?executive_producer .
}
```
```

| | Acc. | Type | Cartesian product | Property direction | Wrong property / type | Entity reference |
|---|---|---|---|---|---|---|
| Full dynamic L2M (0) | 94.6 | 1 | 6 | 5 | 3 | 5 |
| − Delta highlights (a) | 93.8 | 2 | 8 | 4 | | 6 |
| − Decomposition hints (b) | 92.4 | | 14 | | 1 | 5 |
| − Question grounding (c) | 79.2 | 12 | 3 | 1 | 1 | 3 |
| BoW exemplars (d) | 92.2 | 2 | 4 | 9 | 1 | 4 |
| Constant exemplars (e) | 79.9 | | 2 | 7 | 10 | 1 |
| 50% decomposition (f1) | 88.8 | 12 | 3 | 2 | | 3 |
| 2-step decomposition (f2) | 83.2 | 9 | 10 | | | 1 |
| No decomposition (f3) | 75.2 | 9 | 9 | 1 | | 1 |

Table 5: The first column is accuracy for various 1-step ablations of dynamic least-to-most prompting on CFQ, measured on a 500-example subset of MCD1 test set. The other columns show error analysis. For the top row, we look at 20 random samples where the model was incorrect, assigning each to a category. The other rows are similar, except measured on errors introduced by the ablation.

E.2   DYNAMIC LEAST-TO-MOST PROMPTING ON CFQ: ABLATIONS AND ERROR ANALYSIS

Here we include 1-step ablations of the most important components of the dynamic least-to-most method (explained in detail in Appendix A.3). These components are:

- **Delta highlights** are textual hints (such as "# Extend the answer above. Add: ..." and "# Copy the answer from above.") that highlight the change between subsequent subquestions.

- **Decomposition hints** are parenthesis that we add to the natural language questions to indicate the syntactic parse.

- **Question grounding** is a constant prompt prefix that illustrates to the model how to translate the most basic subquestions.

- **Exemplar selection**. *Bag of words (BoW) exemplar selection* is an alternative method to select relevant exemplars. Instead of using the parse tree, it aims to cover with exemplars as many words as possible of the target question. *Constant exemplars* uses a static set of exemplars for all questions (i.e., no dynamically selected exemplars).

- **Decomposition granularity**. *No decomposition* does not decompose the question at all, which means that we then use standard prompting rather than least-to-most prompting. *2-step decomposition* decomposes the question into at most 2 steps, while *50% decomposition* keeps half the decomposition steps of the most granular decomposition.

The second column of Table 5 shows the accuracies of these ablations on a 500-example subset of MCD1. It turns out that ablating delta highlights (93.8% accuracy) and decomposition hints (92.4% accuracy) only lead to a small regression. This matches our intuition because these components cause a relatively small change to the overall prompt. Ablating question grounding on the other hand leads to a much larger regression (79.2% accuracy). The reason for this is that the partial exemplars turn out to be important for the model to correctly translate the first subquestion. The performance of the other ablations are discussed in Section 6.

In addition to the accuracies, Table 5 also contains a qualitative error analysis. For the baseline (first row), we analyzed 20 random errors. For each of the ablations, we analyzed 20 errors that were caused by the ablation (i.e., the examples were correct in the baseline). We make the following observations:

- Delta highlights (a) and BoW exemplar selection (d) have a similar error profile as the baseline. Property direction is a bit more prominent for BoW exemplar selection, which indicates that we do not always retrieve exemplars that illustrate the property in as much detail as in the baseline.

- Decomposition hints (b) mostly help with cartesian products, which makes sense because they make it easier for the model to understand the grouping and connections between the various words of a question.

- Question grounding (c) helpw a lot with types, which is the primary reason why we added them.

- Using constant exemplars (e) leads to a lot of issues with wrong properties/types and property direction. This makes sense since the constant exemplars cannot illustrate all properties and types that are used in the dataset.

- Reducing the decomposition granularity (f1, f2, f3) mostly leads to problems with types and cartesian products. Especially the issues with cartesian products are to be expected because decomposition allows the model to unfold those incrementally.

Below, we provide a few examples to illustrate the different error buckets.

**Unnecessary or missing types**

**Question:** What Dutch spouse of M2 was M0 's employee and founder
**Answer:** SELECT DISTINCT ?x0 WHERE { **?x0 a person** . ?x0 married_to M2 . ?x0 founded M0 . ?x0 employed_by M0 . ?x0 has_nationality Dutch }
**Golden:** SELECT DISTINCT ?x0 WHERE { ?x0 founded M0 . ?x0 employed_by M0 . ?x0

has_nationality Dutch . ?x0 married_to M2}

**Question:** Who was a writer , costume designer , editor , and star of M1 and M2
**Answer:** SELECT DISTINCT ?x0 WHERE { ?x0 starred_in M1 . ?x0 starred_in M2 . ?x0 wrote M1 . ?x0 wrote M2 . ?x0 edited M1 . ?x0 edited M2 . ?x0 costume_designed M1 . ?x0 costume_designed M2 }
**Golden:** SELECT DISTINCT ?x0 WHERE { **?x0 a person** . ?x0 starred_in M1 . ?x0 starred_in M2 . ?x0 edited M1 . ?x0 edited M2 . ?x0 costume_designed M1 . ?x0 costume_designed M2 . ?x0 wrote M1 . ?x0 wrote M2 }

### Incorrect cartesian product

**Question:** Did M1 's costume designer , cinematographer , director , art director , producer , and editor produce , write , and edit M0
**Answer:** SELECT count(*) WHERE { ?x0 directed M1 . ?x0 executive_produced M0 . ?x0 edited M0 . ?x0 wrote M0 . ?x0 cinematographer_of M1 . ?x0 costume_designed M1 . ?x0 produced M1 }
**Golden:** SELECT count(*) WHERE { ?x0 cinematographer_of M1 . **?x0 directed M0** . ?x0 directed M1 . ?x0 edited M0 . ?x0 costume_designed M1 . ?x0 executive_produced M0 . ?x0 produced M1 . ?x0 wrote M0 }

**Question:** Did M2 's editor , director , star , art director , and cinematographer produce , direct , and edit M0 and M1
**Answer:** SELECT count(*) WHERE { ?x0 edited M0 . ?x0 edited M1 . ?x0 directed M0 . ?x0 directed M1 . ?x0 produced M0 . ?x0 produced M1 . ?x0 starred_in M2 . ?x0 art_directed M2 . ?x0 cinematographer_of M2 }
**Golden:** SELECT count(*) WHERE { ?x0 starred_in M2 . ?x0 cinematographer_of M2 . ?x0 directed M0 . ?x0 directed M1 . **?x0 directed M2** . ?x0 edited M0 . ?x0 edited M1 . **?x0 edited M2** . ?x0 art_directed M2 . ?x0 produced M0 . ?x0 produced M1 }

### Wrong property direction

**Question:** Which actor that a cinematographer was influenced by was M1 's spouse
**Answer:** SELECT DISTINCT ?x0 WHERE { ?x0 a actor . ?x0 married_to M1 . **?x0 influenced_by ?x1** . ?x1 a cinematographer }
**Golden:** SELECT DISTINCT ?x0 WHERE { ?x0 a actor . **?x0 influenced ?x1** . ?x0 married_to M1 . ?x1 a cinematographer }

**Question:** What prequel of M0 was executive produced , written , directed , edited , and produced by M1
**Answer:** SELECT DISTINCT ?x0 WHERE { **?x0 has_prequel M0** . ?x0 executive_produced_by M1 . ?x0 produced_by M1 . ?x0 edited_by M1 . ?x0 directed_by M1 . ?x0 written_by M1 }
**Golden:** SELECT DISTINCT ?x0 WHERE { ?x0 directed_by M1 . ?x0 edited_by M1 . ?x0 executive_produced_by M1 . ?x0 produced_by M1 . **?x0 has_sequel M0** . ?x0 written_by M1 }

### Wrong property / type

**Question:** Was a film director 's child M0 's cinematographer
**Answer:** SELECT count(*) WHERE { ?x0 child_of ?x1 . ?x1 a film_director . **?x0 cinematographed M0** }
**Golden:** SELECT count(*) WHERE { **?x0 cinematographer_of M0** . ?x0 child_of ?x1 . ?x1 a film_director }

**Question:** Was a company whose employee wrote M2 M1 's employer
**Answer:** SELECT count(*) WHERE { **?x0 a company** . ?x0 employed M1 . ?x0 employed ?x1 . ?x1 wrote M2 }
**Golden:** SELECT count(*) WHERE { **?x0 a employer** . ?x0 employed ?x1 . ?x0 employed M1 . ?x1 wrote M2 }

**Mixing up entity references**

**Question:** What costume designer was a film director 's Canadian female parent
**Answer:** SELECT DISTINCT ?x0 WHERE { ?x0 a costume_designer . **?x0 a film_director** . ?x0 has_gender female . ?x0 has_nationality Canadian . ?x0 parent_of ?x1 } }
**Golden:** SELECT DISTINCT ?x0 WHERE { ?x0 a costume_designer . ?x0 parent_of ?x1 . ?x0 has_gender female . ?x0 has_nationality Canadian . **?x1 a film_director** }

**Question:** Did a male actor whose spouse married M2 play M0
**Answer:** SELECT count(*) WHERE { ?x0 portrayed M0 . ?x0 married_to ?x1 . ?x1 married_to M2 . ?x1 a actor . **?x1 has_gender male** }
**Golden:** SELECT count(*) WHERE { ?x0 portrayed M0 . **?x0 has_gender male** . ?x0 married_to ?x1 . ?x1 a actor . ?x1 married_to M2 }

### E.3 FAIR COMPARISON AGAINST OTHER PROMPTING TECHNIQUES

In the comparison in Figure 5, vanilla few-shot and chain-of-thought prompting have an advantage over least-to-most prompting because we sample multiple exemplar lists ($n = 4$) and multiple outputs per list ($s = 4$) using temperature-based decoding. This yields $n \cdot s = 16$ outputs per input, which are aggregated using self-consistency. When using $n = 1$ and $s = 1$ with greedy decoding, the comparison between these prompting techniques and dynamic least-to-most is more fair, and the benefits of dynamic least-to-most are more prominent. Chain-of-thought achieves 75.4% accuracy (down from 87.2%), and vanilla few-shot achieves 69.8% accuracy (down from 80.8%). Dynamic least-to-most substantially outperforms both of these without using self-consistency, and achieves 94.6% on the same 500-sentence subset of MCD1 validation data.

## F REPRODUCIBILITY SUMMARY

At all points in this work, we aim to make our methods and experiments easily reproducible, and we hope our findings will have impact in part through others using the same or similar methods for new tasks. Here we summarize critical components of our work and where their relevant details are described:

- Main Prompts: In lieu of code, we provide the exact prompts that we executed to obtain model predictions. Prompts for CFQ: syntactic parsing (Appendix A.1), dynamic least-to-most (Appendix A.3). Prompts for COGS: syntactic parsing (Appendix B.1), dynamic least-to-most (Appendix B.3). Exemplars are chosen using the methods described in Section 3.2, and Appendix A.2 (for CFQ) and Appendix B.2 (for COGS). The prompts include the exemplars found from a concrete input.

- Additional Prompts: We only use vanilla few-shot and chain-of-thought to compare against dynamic least-to-most on CFQ. The chain-of-thought prompt used to bootstrap rationale is in Appendix C.1, and the one used for evaluation data is in Appendix C.2. The prompt includes the exemplars found from a concrete input. The vanilla few-shot prompt is similar, but all examples use the vanilla input/output format. Exemplars are chosen using the methods described in Section 4.2.

- Dataset Preparation and Evaluation: We performed both pre-processing and a post-processing output normalization step in order to make semantic parsing with freebase identifiers more feasible for prompting. Described in Section 5.1, Appendix D.1.1, D.1.2, D.2.

- Exemplar Pool: Methodology for constructing exemplar pools is described in Section 3.2.

- Hyperparameters: We didn't require any extensive hyperparameter search. The hyperparameters we used are described in Section 5.2 and Appendix D.3.

