# OpenReview forum: "Compositional Semantic Parsing with Large Language Models"
_ICLR.cc/2023/Conference — ICLR 2023 poster_

### Official Review · Reviewer_e27M · 2022-10-21

**Confidence:** 3
**Correctness:** 3
**Technical Novelty And Significance:** 3
**Empirical Novelty And Significance:** 3
**Recommendation:** 8

**Clarity, Quality, Novelty And Reproducibility:**

The paper is quite clear but many important details are in the appendix due to the complexity of prompts and heuristics.

This extension to least-to-most prompting is completely novel, to the best of my knowledge.

**Strength And Weaknesses:**

Strength: this paper introduces a novel extension to least-to-most prompting with several improvements: tree decomposition; example selection to fit more relevant examples into the prompt (in realistic cases, the label space is too large to be able to fit examples for everything); subproblems are solved by giving context to the model. The results shown on CFQ and COGS are very good.

Weakness (addressed in rebuttal): The approach requires dataset-specific prompts (for syntactic parsing) and heuristics (for exemplar selection). It is unclear how much the final performance depends on the details of the prompts/heuristics and how much the reported results overfit to these choices. Also, it looks like a lot of manual effort is needed if one wants to apply this method to a new dataset.

**Summary Of The Paper:**

This paper extends the least-to-most prompting technique to solve more realistic semantic parsing tasks. Specifically, the approach works as follows:

1. Prompt a large language model with examples to perform syntactic parsing of the input sentence and decompose it into increasingly simpler parts
2. Select relevant examples based on the decomposition obtained in 1
3. Prompt the large language model with the relevant examples obtained in 2 in order to solve the task on all parts of the input (according to the decomposition found in 1), starting from simple parts and ending with the full input sentence.

The paper shows positive results on the CFQ and COGS datasets while only selecting examples from a small subset of the training data (e.g., 1%).

**Summary Of The Review:**

I believe this paper is overall worthy of publication thanks to novelty and good results.

---

> ### Author Response · Authors · 2022-11-17
> **Author Response**
>
> Thanks for the constructive feedback!
>
> **Overfitting of prompts/heuristics**
>
> > It is unclear how much the final performance depends on the details of the prompts/heuristics.
>
> Thanks for this important question. In the revision, we added an extensive set of ablations that should address this question (Table 3 and Appendix E). We also describe the ablations in detail in our response to reviewer WSSM.
>
> > It is unclear how much the reported results overfit to these choices.
>
> We would like to note that for both datasets, the size of the prompt is a tiny fraction of the size of the test set (which is about 12k examples for CFQ). We informed our design choices for prompts and heuristics by evaluating on only **100 validation examples**. Finally, once our approach was stable, then we evaluated against the test data.
> Considering this setup, we do not share the concern of overfitting. On the contrary, we would like to point out that more traditional approaches based on end-to-end supervised learning would typically use a training set that is larger than the test set, which elevates risk of overfitting compared to our approach. Also, note that we used the same prompts/heuristics for all MCD splits, which speaks against overfitting.
>
> **Cost of prompt design vs. data collection**
>
> > a lot of manual effort is needed if one wants to apply this method to a new dataset
>
> Instead of focusing on the manual effort needed to apply this method to a **new dataset**, we encourage the reviewer to consider the effort needed to apply it to a **new problem** (which initially comes without training data).
>
> Indeed, a significant part of the motivation for using LMs is the fact that many of the existing approaches based on supervised learning are expensive to apply to real-world problems because of the lack of training data. While training data is a given for most benchmarks, it’s common for machine learning practitioners to attempt new real-world tasks where training data is not initially available. This typically involves an expensive manual effort to annotate new training data.
>
> When comparing the **total manual effort** when applying a method to a **new problem** (training set creation + prompt design), we believe that our approach compares favorably to alternative approaches such as LEAR. For CFQ, we were able to exceed the accuracy of the previous state of the art (LEAR) using 500x fewer training examples (see Figure 5, which shows that 200 exemplars are enough for exceeding 91% accuracy). Regarding the manual effort required for this problem, this means that we needed to write 200 training examples instead of 3 times 95k training examples (for the 3 MCD splits). Compared to this, the manual effort for writing the prompt is minor. Indeed, the total number of ASCII characters used for our prompts is over 1000x smaller than the additional training data used by traditional approaches based on supervised learning.
>
> **Complexity of prompt design**
>
> > The paper is quite clear but many important details are in the appendix due to the complexity of prompts and heuristics.
>
> To make more clear which are the important aspects of our prompt design we performed new ablations about exemplar selection and question decomposition. These are in our response to reviewer WSSM and in our new revision (Table 3 and Appendix E).
>
> > It is unclear how much the final performance depends on the details of the prompts/heuristics
>
> This is also answered by our new ablations (Table 3 and Appendix E).

---

> > ### Comment · Reviewer_e27M · 2022-11-21
> > **Review updated**
> >
> > Thanks for the reply and the additions to the paper, you made good points and I think the new ablations are very useful. I updated my review and score.

---

### Official Review · Reviewer_WSSM · 2022-10-21

**Confidence:** 3
**Correctness:** 3
**Technical Novelty And Significance:** 2
**Empirical Novelty And Significance:** 3
**Recommendation:** 5

**Clarity, Quality, Novelty And Reproducibility:**

The paper is nice to read. It argues well about the motivation and practical difficulties to apply least-to-most prompt tuning method to more complex semantic parsing tasks and well address the concerns. Both the motivation and the solution is clearly exposed.

The paper is an application of least-to-most prompt tuning applied to compositionality and the techniques proposed to address the concerns are quite standard NLP techniques. I don't rate highly on originality especially there misses a thorough ablation studies showing why such technical choices are important.

**Strength And Weaknesses:**

Strength:
1. The paper has clear motivations to achieve compositionality for more real-life settings
2. The proposed method addresses practical concerns
3. The method shows great performance in two real-life like datasets (compared to SCAN)

Weakness:
1. The paper consists mainly an application of least-to-most prompt tuning method with relatively standard NLP techniques
2. The ablations studies of the paper's contribution (sentence decomposition and dynamic prompt selection is missing from the main paper). The question that leave readers are for example: what is an effective sentence decomposition? what is the performance with no setntence decomposition but just some dynamic prompting? What about simpler prompt selection method?
3. For CFQ dataset, the paper mentions that they use a list of manually chosen examplars (e.g. footnote 3), this limits a bit the applicability of the proposed method.

**Summary Of The Paper:**

Compositionality is an important ability of a model and in this paper, the authors propose to achieve such capacity by using a novel method inspired by least-to-most prompting. It addresses two practical challenges for applying least-to-most prompting:
1. how to decompose the questions to generate least-to-most prompting
2. In practice the uttererances can vary a lot, so how to choose prompt from a large pool

By addressing the above difficulties, the athors propose an s enhanced least-to-most prompt tuning method; empirically, the authors achieve SOTA/near SOTA performance for two popular compositionality bench marks COGS and CFQ.

**Summary Of The Review:**

The paper tackles how to achieve semantic parsing compositionality using least-to-most prompt tuning. It addresses two practical challenges that are 1) how to decompose a sentence in this case 2) how to dynamically choose the best examplars. The paper gives clear motivation and intuition about the proposed methods and achieve convincing results for two challenging complex semantic parsing tasks focusing on compositionality.

On the downside, the paper doesn't study closely for its technical choices, the decomposition alone is quite complicated (i.e. Appendix A.1 and B.1) and the paper doesn't mention how important and why such choices matter for the end performance; I don't see detailed ablation analysis for examplar choosing algorithms either and there are limitations of the methods shown by the implementation (e.g. manually choosing some examplars).

PS: I have read the authors' comments, thanks a lot for authors to give detailed comments as well as new experimental results. My two main concerns (how to choose examples and ablation studies about the decomposition) have been answered in detail. The paper clearly has its own merits; however, I found the paper doesn't dive deep enough in its decomposition choices (e.g. Reviewer P921 mentioned a possiblity to apply syntactic parsing to do decomposition) and I am convinced that the manual step can be applied to other tasks but is also incurs cost so not sure if such appraoches will be easily widely adopted. I would not change my score although I sincerely appreciate the informative and constructive discussions.

---

> ### Author Response · Authors · 2022-11-17
> **Author Response**
>
> We’d like to thank the reviewer for their very informative feedback.
>
> **W2: Missing ablations**
>
> We’ve run new ablations in response to the following comment:
>
> > The question that leaves readers are for example: What is an effective sentence decomposition? What is the performance with no sentence decomposition but just some dynamic prompting? What about simpler prompt selection methods?
>
> These new ablations are in our new revision (Table 3) and we believe greatly strengthen our work. We measure performance of various settings on a 500-example subset of CFQ MCD1 test data. The full dynamic least-to-most method outperforms all ablations at 94.6% accuracy, and we describe the new ablations below.
>
> **(on exemplar selection)** The ablations assist in understanding the robustness of dynamic least-to-most to design choices. For instance, bag-of-words selection scores 92.2%, indicating the benefit of our top-down/bottom-up search procedure. Using a random fixed set of examples scores 76.4%, indicating more generally the importance of informed exemplar selection for complex tasks like CFQ.
>
> **(on decomposition)** Performance decreases when using simpler decomposition with half as many steps (88.8%), at most two steps (83.2%), or no decomposition at all (75.2%). This matches the intuition behind least-to-most, that reducing a complex problem into relatively easier subproblems is beneficial for compositional generalization.
>
> **(additional ablations and error analysis)** We ran other new ablations that do not easily fall into the above categories. For example, performance decreases when we do not add parentheses to indicate the syntactic parse of the sentence (92.4%). This reinforces the benefits of our tree-structured approach.
>
> We also added a qualitative error analysis for our best approach as well as for the new errors introduced by each ablation (Appendix E). This analysis shows for example that removing parentheses to indicate the syntactic parse typically leads to additional errors related to cartesian products, and reducing the decomposition granularity typically leads to additional errors related to cartesian product and types.
>
> We address the remaining concerns below:
>
> **W1: The work is just an application of least-to-most prompting**
>
> > The paper consists mainly an application of least-to-most prompt tuning method with relatively standard NLP techniques
>
> We kindly disagree with this claim and we explain in section 2.3 the difficulties when applying least-to-most prompting unchanged to tasks like CFQ. It is reinforced in our new ablations that it is necessary to refine least-to-most. Specifically, our best performance is from selecting exemplars based on the decomposition tree. Both this method of exemplar selection and the decomposition (which relies on syntactic parsing) are novel techniques in prompting.
>
> **W3: Manual example selection is limiting**
>
> > For CFQ dataset, the paper mentions that they use a list of manually chosen examplars (e.g. footnote 3), this limits a bit the applicability of the proposed method.
>
> The footnote that you are referring to is not very clear, which led to a misunderstanding here (we updated the footnote in the latest version of the paper). Our best approach for CFQ does not use manually selected exemplars. Instead, it is dynamically selects exemplars from a random subset of 1000 training examples. What the footnote is referring to are a set of manually written and generic partial questions that teach the model basic structure associated with the basic question types (e.g., “Did” vs. “W-Question”). As we show in our new ablation and error analysis (Appendix E), this mostly helps the model to deal with types.
>
> We do not think that this limits the applicability of the proposed method. Instead, we believe that this is further evidence that our method is much more than “just an application of least-to-most prompting”. It shows that adding some additional instruction can significantly improve the performance. While it is true that this requires some additional work, we do not see a reason why this technique would not be applicable to other datasets. Also, note that we did not need to use such additional instructions for COGS.

---

### Official Review · Reviewer_P921 · 2022-10-24

**Confidence:** 2
**Correctness:** 4
**Technical Novelty And Significance:** 3
**Empirical Novelty And Significance:** 3
**Recommendation:** 6

**Clarity, Quality, Novelty And Reproducibility:**

Clarity: the paper is well written.

Quality: good.

Novelty: the idea is new, and the experiment results are impressive.

Reproducibility: good.

**Strength And Weaknesses:**

Strength:

1. The idea used seems new and interesting.
2. The experiment results are very good and impressive.

Weakness:
1. I suggest the authors also include the baseline results using the same number of training data as the proposed methods for a better comparison if possible.
2. For the syntactic parsing part, although the experiment results of the proposed method is very good, it is not clear why we should use LM to parse instead of the more "classical" ways like training a PCFG like model. In fact, the paper from Qiu et al. uses the QCFG model to "parse". Is LM perform better? or it is much more sample-efficient? or it is much simpler?

**Summary Of The Paper:**

This paper considers compositional generalization and proposes a "dynamic least-to-most prompting" method to tackle the problem. "dynamic least-to-most prompting" consists of three steps: decomposition using syntactic parsing, dynamic exemplar selection, and sequential solution. The experiment results are impressive: on CFQ dataset, the proposed algorithm beats all the existing baselines while using much less training data. On COGS, although the performance is a little bit lower than one of the baseline, the number of training data used is much less.

**Summary Of The Review:**

I am not very familiar with this sub-field, and thus based on my evaluation, I think this paper is interesting and make some contribution to compositional generalization. The paper is well written and the comparison with the previous work seems enough to me. Besides, the numerical experiments show the effectiveness of the proposed method.

---

> ### Author Response · Authors · 2022-11-17
> **Author Response**
>
> Thank you for the thoughtful feedback.
>
> **Could you compare baselines using the same number of training data?**
>
> > I suggest the authors also include the baseline results using the same number of training data as the proposed methods for a better comparison if possible.
>
> Thanks for bringing this up! We compare our approach with fully supervised baselines that use the full training data, and we only use a fraction of the data (about 1% for CFQ) when selecting exemplars. It is reasonable to assume that the baselines would do **worse** when using less training data.
>
> To verify this, we have run a new experiment with the previous state of the art model for CFQ, LeAR (Liu et al., 2021). We limit the amount of training data of LeAR to 1000 random samples from MCD1 train. On the test set, **LeAR only achieves 84.60% accuracy when trained with 1000 samples**, which is significantly worse than the LeAR trained with the full data (91.7%) and also significantly worse than our dynamic least-to-most approach that only uses 1000 train samples in the exemplar pool (94.3%).
>
> There are some important details for attaining good results with LeAR.
>
> a) For LeAR we need to use curriculum learning by training on short sentences first, then extending to the full data. Without curriculum learning, *LeAR only achieves 40-50% accuracy*. (see sec 4.3 in Liu et al. for more details)
>
> b) LeAR heavily depends on an automatically learned alignment lexicon between input and output tokens (also called “phrase table” in Liu et al.). This lexicon is learned using GIZA++. We use the same lexicon from the original LeAR paper, which is derived from the full train data and gives an unrealistic advantage in the lower data setting. For this reason, the 84.60% accuracy should be treated as an “upper bound” on LeAR performance, and *it is likely a lexicon derived from only 1000 samples would have resulted in catastrophically worse accuracy*. For more details on lexicon usage in LeAR, please see sec 4.2 and appendix D in Liu et al.
>
> Also, anecdotally, earlier in our research we ran T5 baselines trained only on 1000 train samples resulting in substantially lower accuracy (10-20 point drop).
>
> These new results clearly show CFQ is even more challenging in the low data setting, so it is especially notable that our dynamic least-to-most approach outperforms baselines that were trained with the full training data.
>
> Chenyao Liu, Shengnan An, Zeqi Lin, Qian Liu, Bei Chen, Jian-Guang Lou, Lijie Wen, Nanning Zheng, Dongmei Zhang. Learning Algebraic Recombination for Compositional Generalization. 2021.
>
> **Would you discuss advantages over grammar based semantic parsers?**
>
> > For the syntactic parsing part, although the experiment results of the proposed method is very good, it is not clear why we should use LM to parse instead of the more "classical" ways like training a PCFG like model. In fact, the paper from Qiu et al. uses the QCFG model to "parse". Is LM perform better? or it is much more sample-efficient? or it is much simpler?
>
> By using  a LM for parsing we can easily derive a syntactic parser efficiently with only a few exemplars, and without needing a large amount of training data. More generally, the flexibility of the LM is useful for semantic parsing across a variety of tasks and domains. As Qiu et al. 2022 emphasize in Appendix C.5, their approach **does not extend to CFQ** because of the limitations in quasi-synchronous grammars which they rely on.
>
> One additional benefit is we can easily control for syntactic parse tree granularity in our approach; we did not explore this extensively, but it was relatively low effort to use the LM for syntactic parsing rather than integrate an external parser.
>
> Linlu Qiu, Peter Shaw, Panupong Pasupat, Pawel Nowak, Tal Linzen, Fei Sha, and Kristina Toutanova. Improving compositional generalization with latent structure and data augmentation. 2022.
>
> **Why don’t you use an external parser for the syntactic parsing part?**
>
> Our paper proposes a novel and data efficient framework for handling compositional semantic parsing problems. If we had used external syntactic parser, we couldn't confidently claim data efficient learning because the external parser already learned/knew about the parsing task, possibly by using techniques that are not as data efficient such as end-to-end fully supervised learning.

---

### Official Review · Reviewer_gAic · 2022-10-25

**Confidence:** 3
**Correctness:** 4
**Technical Novelty And Significance:** 4
**Empirical Novelty And Significance:** 4
**Recommendation:** 8

**Clarity, Quality, Novelty And Reproducibility:**

* CFQ in the abstract is used before introduced.

**Strength And Weaknesses:**

S:
* The paper is well-written, well-motivated, and easy to follow.
* The Dynamic least-to-most prompting is a good idea that it first sequentially predicts solutions to subproblems before generating the final output, where the subproblems are extracted through different prompts.

W:
*There is no obvious weakness in this work, given the improvement and effectiveness of 1% data is very promising.
* One minor thing could be there is no analysis between task complexity (for example, number of tokens, number of vocabulary, number of decomposed subquestions) and performance. Another minor thing is that there is no study on how sensitive the Codex model is to the data preprocessing used in the work as shown in the Appendix.



**Summary Of The Paper:**

The authors first perform decomposition by dividing the syntactic parsing task into multiple steps and provide LMs with exemplars that illustrate the task to be performed. They then choose an exemplar pool with 1000 examples on CFQ and 89 for COGS. And they select for each input between 4 and 35 exemplars for CFQ and between 1 and 3 exemplars for COGS. They highlight the effectiveness of such compositional ability and set new SOTA results for CFQ while requiring only 1% of the training data used by traditional finetuning approaches.

**Summary Of The Review:**

This paper is working on a natural language semantic parsing benchmark, and the authors provide a dynamic version of least-to-most prompting to show promising generalization ability.

---

> ### Author Response · Authors · 2022-11-17
> **Author Response**
>
> Thank you for your thoughtful comments.
>
> > analysis between task complexity
>
> Our approach is effective on two different tasks, CFQ and COGS. COGS is notable as it has a much larger vocabulary than CFQ. The COGS vocab size for source and target is 743 and 658, compared with 96 and 96 for CFQ.
>
> > how sensitive the Codex model is to the data preprocessing
>
> It was important that we replaced property names in CFQ with human-readable ones (see footnote 4), this preprocessing is increasingly standard for LM evaluation (see Herzig et al. 2021). This was a small effort, and we simply aimed to be consistent in our naming (e.g. using similar tenses for verbs). For COGS we simply used the preprocessing from Qiu et al. 2022 (sec 5.1).
>
> Jonathan Herzig, Peter Shaw, Ming-Wei Chang, Kelvin Guu, Panupong Pasupat, and Yuan Zhang. Unlocking compositional generalization in pre-trained models using intermediate representations. 2021.
> Linlu Qiu, Peter Shaw, Panupong Pasupat, Pawel Nowak, Tal Linzen, Fei Sha, and Kristina Toutanova. Improving compositional generalization with latent structure and data augmentation. 2022.

---

### Author Response · Authors · 2022-11-18
**Thank you to all the reviewers**

We appreciate the valuable feedback.

### Review Summary

Here are the **main strengths** mentioned across reviews:

**Well Written**

* (gAIc) The paper is well-written, well-motivated, and easy to follow.

**Well Motivated**

* (gAIc) The Dynamic least-to-most prompting is a good idea that it first sequentially predicts solutions to subproblems before generating the final output, where the subproblems are extracted through different prompts.
* (WSSM) The paper has clear motivations to achieve compositionality for more real-life settings

**Practical**

* (WSSM) The proposed method addresses practical concerns

**Novel**

* (P921) The idea used seems new and interesting.
* (e27M) This paper introduces a novel extension to least-to-most prompting with several improvements: tree decomposition; example selection to fit more relevant examples into the prompt (in realistic cases, the label space is too large to be able to fit examples for everything); subproblems are solved by giving context to the model.

**Strong Results**

* (P921) The experiment results are very good and impressive.
* (WSSM) The method shows great performance in two real-life like datasets (compared to SCAN)
* (e27M) The results shown on CFQ and COGS are very good.

### Additional Experiments

We've ran new experiments based on the reviews, which we believe strengthens and increases the clarity of our work:

**New Ablations**

We've introduced new ablations to understand the influence of exemplar selection and sentence decomposition in dynamic least-to-most. The results are in this table, and measure accuracy on a 500-sample subset of the CFQ MCD1 test set. Further discussion is in our other response and in the newly revised paper in Section 6 and Table 5.

|                          | MCD1 |
|--------------------------|------|
| Dynamic Least-to-Most    | 94.6 |
| Using BoW Exemplars      | 92.2 |
| Using Constant Exemplars | 76.4 |
| 50% Decomposition        | 88.8 |
| 2-Step Decomposition     | 83.2 |
| No Decomposition         | 75.2 |

In short, these ablations show the benefit of our top-down/bottom-up exemplar selection algorithm as well as the benefit of multi-step decomposition.

Further ablations, error analysis, and extensive discussion are have been added in Appendix E.2.

**New Baseline**

For a more comparison between our approach and baseline methods, we trained LeAR using only 1000 samples, which is equal to the size of our exemplar pool. We report the accuracy of the full MCD1 test set of CFQ below.

|         | Train Samples                 | MCD1 |
|--------------------------|------|---|
| Dynamic Least-to-Most | 1,000    | 94.3 |
| LeAR | 95,743    | 91.7 |
| LeAR | 1,000 | 84.6|
| LeAR w/o Curriculum Learning | 1,000 | <50.0 |

It's worth noting that LeAR depends on curriculum learning for good performance. More importantly, LeAR depends on a lexicon mapping between input-output tokens that is learned automatically from the full training data. Even though we train with 1000 samples, we use the mapping from the full data. So these results act as an upperbound on LeAR performance, and without this mapping, performance would likely be much worse.

The new baseline results reinforce the benefit of our dynamic least-to-most approach when only a small amount of training data is available.

---

### Decision · Program_Chairs · 2023-01-20

**Decision:**

Accept: poster

**Justification For Why Not Higher Score:**

The proposed method is an extension of the least-to-most prompting method to more complex dataset.  The novelty is a bit small.

**Justification For Why Not Lower Score:**

The result is impressive: using only 1% training data to reach a better performance.

**Metareview: Summary, Strengths And Weaknesses:**

Summary:

Compositionality is a traditional and important NLP problem.  This paper proposed a method to extend the least-to-most prompting method for large language models (LLMs) to solve more complex compositional problems (CFQ and COGS ) then existing work (SCAN).  Firstly, decompose the questions into increasingly simpler parts using syntactic parsing by prompting an LLM with examples; Secondly, select relevant examples based on the decomposition; Thirdly, solve the problem from simple parts to their combination by prompting the LLM with the relative examples.  Experiments show the proposed method outperforms baselines by selecting examples from a subset of the training data which is much smaller (1%).

Strengths:

The idea of dynamic is least-to-most prompting novel and interesting.
The results on CFQ and COGS are good and impressive.
The paper is well-motivated, clearly written, and easy to follow.

Weaknesses:

Lack of ablation studies.  For parsing, it would be good to compare the proposed method with traditional paring methods like PCFG.  The dataset is quite specific and there are some euristic manual work.  It is not known if it is easy to transfer to a new dataset.


**Note From Pc:**

if the above contains the word "oral" or "spotlight" please see: "oral" presentation means -> notable-top-5% and "spotlight" means -> notable-top-25%. As stated in our emails, we are disassociating presentation type from AC recommendations

**Summary Of Ac-Reviewer Meeting:**

NA